# Long-term implicit memory for sequential auditory patterns in humans

**Roberta Bianco[1]\*, Peter MC Harrison[2], Mingyue Hu[1], Cora Bolger[1], Samantha Picken[1], Marcus T Pearce[2,3], Maria Chait[1]**

[1]UCL Ear Institute, University College London, London, United Kingdom; [2]School of Electronic Engineering and Computer Science, Queen Mary University of London, London, United Kingdom; [3]Department of Clinical Medicine, Aarhus University, Aarhus, Denmark

**Abstract** Memory, on multiple timescales, is critical to our ability to discover the structure of our surroundings, and efficiently interact with the environment. We combined behavioural manipulation and modelling to investigate the dynamics of memory formation for rarely reoccurring acoustic patterns. In a series of experiments, participants detected the emergence of regularly repeating patterns within rapid tone-pip sequences. Unbeknownst to them, a few patterns reoccurred every ~3 min. All sequences consisted of the same 20 frequencies and were distinguishable only by the order of tone-pips. Despite this, reoccurring patterns were associated with a rapidly growing detection-time advantage over novel patterns. This effect was implicit, robust to interference, and persisted for 7 weeks. The results implicate an interplay between short (a few seconds) and long-term (over many minutes) integration in memory formation and demonstrate the remarkable sensitivity of the human auditory system to sporadically reoccurring structure within the acoustic environment.

## Introduction

Memory is a crucial component of sensory perception, on multiple processing levels (*Bale et al., 2017*; *Muckli and Petro, 2017*). In the auditory modality, the ability to identify essentially any sound source, from footsteps to musical melody, requires the capacity to hold consecutive events in memory so as to link past and incoming information into a coherent emerging representation (*Koelsch et al., 2019*; *McDermott et al., 2013*; *Winkler et al., 2009*). Whilst traditional models of sensory memory (e.g. *Cowan, 1998*) argued that such sensory traces are characterized by short retention times and computational encapsulation, a large body of work has since revealed that observers can retain detailed sensory information implicitly, over long periods (*Arciuli and Simpson, 2012*; *Chun, 2000*; *Jiang et al., 2005*; *Kim et al., 2009*; *Vogt and Magnussen, 2007*; *Winkler and Cowan, 2005*). A compelling instance was demonstrated by *Agus et al., 2010*; (see also *Agus and Pressnitzer, 2013*; *Kang et al., 2017* who showed that naive listeners readily remembered certain spectro-temporal features of random noise bursts, such that reoccurring snippets were recognized weeks after initial exposure.

Here, we focus on long-term memory formation for arbitrary frequency patterns within rapidly unfolding sequences of discrete sounds. We ask whether naïve listeners can become sensitized to sparsely reoccurring tone sequences and investigate the conditions under which such memories are formed. To formalize the underlying psychological mechanisms, we simulate human performance with a probabilistic model of sequential memory (*Harrison et al., 2020*; *Pearce, 2018*).

The experimental design (*Figure 1*) capitalizes on a paradigm developed by *Barascud et al., 2016* for measuring listeners' sensitivity to complex acoustic patterns. Using fast sequences of short tones, they showed that listeners can rapidly detect the transition to a regularly repeating pattern

**\*For correspondence:** r.bianco@ucl.ac.uk

**Competing interests:** The authors declare that no competing interests exist.

**eLife digest** Patterns of sound – such as the noise of footsteps approaching or a person speaking – often provide valuable information. To recognize these patterns, our memory must hold each part of the sound sequence long enough to perceive how they fit together. This ability is necessary in many situations: from discriminating between random noises in the woods to understanding language and appreciating music. Memory traces left by each sound are crucial for discovering new patterns and recognizing patterns we have previously encountered. However, it remained unclear whether sounds that reoccur sporadically can stick in our memory, and under what conditions this happens.

To answer this question, Bianco et al. conducted a series of experiments where human volunteers listened to rapid sequences of 20 random tones interspersed with repeated patterns. Participants were asked to press a button as soon as they detected a repeating pattern. Most of the patterns were new but some reoccurred every three minutes or so unbeknownst to the listener.

Bianco et al. found that participants became progressively faster at recognizing a repeated pattern each time it reoccurred, gradually forming an enduring memory which lasted at least seven weeks after the initial training. The volunteers did not recognize these retained patterns in other tests suggesting they were unaware of these memories. This suggests that as well as remembering meaningful sounds, like the melody of a song, people can also unknowingly memorize the complex pattern of arbitrary sounds, including ones they rarely encounter.

These findings provide new insights into how humans discover and recognize sound patterns which could help treat diseases associated with impaired memory and hearing. More studies are needed to understand what exactly happens in the brain as these memories of sound patterns are created, and whether this also happens for other senses and in other species.

(REG) from a sequence of random tones (RAN). Sequences were novel and too rapid to allow for conscious tracking, but on most trials, participants were able to respond soon after the onset of the second cycle of regularity, implicating an efficient memory for the immediate sequence context. Here, we ask how this memory is affected if the tone pattern was already experienced in the past.

Reaction times in *Barascud et al., 2016* were consistent with those obtained from an ideal-observer model based on prediction by partial matching (PPM; *Pearce, 2005*; *Pearce, 2018*). Shown to be an effective model of human auditory sequence learning on multiple time scales (*Agres et al., 2018*; *Di Liberto et al., 2020*; *Harrison and Pearce, 2018*; *Pearce, 2018*; *Pearce and Wiggins, 2006*), this model proposes that listeners acquire an internal representation of the sound input by keeping track of multiple-order Markovian transition probabilities. This context is then used to evaluate the (un)expectedness of ensuing sounds by deriving a measure of surprisal (information content – IC; negative log probability). RAN and REG sequences differ in unexpectedness (high for RAN, low for REG). The transition from a random to a regular pattern (RANREG stimulus) can therefore be detected as a salient drop in information content in the model output (*Figure 1*) which reflects increasing compatibility between the incoming sounds and the stored context. The pattern of behavioural reaction times as well as brain response latencies recorded from naive, passively listening participants (*Barascud et al., 2016*; *Southwell et al., 2017*; *Southwell and Chait, 2018*) suggest that listeners indeed identify the emergence of regularity by detecting the associated drop in information content and that such tracking of instantaneous expectedness constitutes an automatic, inherent aspect of auditory sequence processing.

We used a combination of behavioural manipulation and modelling to examine the durations over which these memory representations are maintained by introducing rare pattern reoccurrences. One might expect that detection of regularities benefits not only from immediate sequence context, but also from traces accumulated over a longer period. Participants listened to RAN and RANREG sequences (as shown in *Figure 1*, see stimulus examples: 'Sound - RAN', and 'Sound - RANREG'), and were instructed to press a keyboard button as soon as possible when a transition to REG was detected. New sequences were generated on each trial, but unbeknownst to participants, a few different regular patterns reoccurred very sparsely (every ~3 min) across trials (RANREGr).

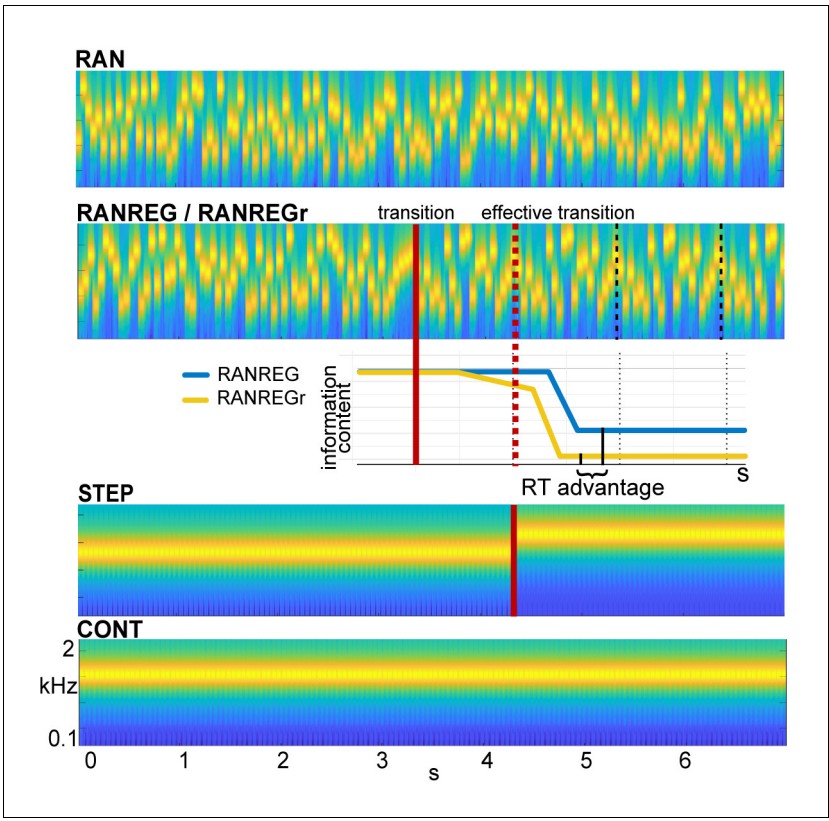

**Figure 1.** Example stimuli. Sequences were generated anew on each trial from a pool of 20 tone-pips of 50 ms duration each. RAN sequences were generated by randomly sampling from the full pool with replacement; RANREG sequences contained a transition from a random (RAN) to a regularly repeating cycles of 20 tone-pips (REG, cycles are marked with dashed lines). Therefore, the transition was manifested as a change in pattern only, whilst maintaining the same long-term first-order statistics. The transition (randomized between 3 and 4 s post onset) is indicated by a red line; the red dashed line marks the 'effective' transition – the point at which the pattern starts repeating and hence becomes statistically detectable. Participants were instructed to respond to such transitions (50% of trials) as soon as possible. STEP stimuli, containing a step change in frequency, (and their 'no change' control, CONT) were also included in the stimulus set for the purpose of estimating simple reaction time. Three (six in Exp. 4 and Exp. S1 in Appendix 1) particular regular patterns (REGr) were presented identically across three trials within a block (RANREGr). Reoccurrences were spaced ~3 min apart. Different REGr were used for each participant. A schematic representation of outputs from the observer model is provided to illustrate how pattern reoccurrence might affect reaction time. For each tone in a sequence, the model outputs information content (IC) as a measure of its unexpectedness, given the preceding context. After the transition from a RAN to REG pattern, the IC drops over a few consecutive tones, reflecting the discovery of the REG. The brain is hypothesized to be sensitive to this change in IC, and once sufficient evidence has been accumulated, the emergent regularity 'pops out' perceptually. Therefore, RTs to onset of regularities can be used to quantify the amount of sensory information (number of tone-pips), required to detect the increasing predictability within the unfolding sequence. The black solid lines indicate the crossing of this putative evidence threshold (when the information content becomes clearly distinguishable from the RAN baseline). For novel patterns (blue line), this typically occurs within the second cycle. For reoccurring patterns (yellow line), IC is expected to show an earlier drop, and therefore lead to faster RT ('RT advantage').

We hypothesized that, if the stored representation of a pattern strengthens through repetition, the information content associated with a transition to a familiar regularity will dip earlier than that associated with a novel regular pattern (*Figure 1*, yellow line in the cartoon model), reaching the putative detection threshold more quickly. Behaviourally, this should be revealed as faster reaction times to reoccurring patterns ('RT advantage' in *Figure 1*). The size of this effect may provide a window into the latent variables associated with the retention of sensory information in memory.

Several properties render this paradigm attractive. First, all sequences consist of the same 20 frequency 'building blocks'. This simplifies parametrization and modelling of the task, while retaining sufficient pattern complexity (there are more than a trillion permutations of 20 frequencies). Second, these 20 frequencies are isochronous and occur with equal probability and roughly equal temporal density in all conditions: stimuli are thus matched in terms of long-term spectrum, average statistics and time patterning. The only difference between RAN and REG patterns and, importantly, between REG and REGr patterns, is the specific arrangement of these tone-pips over time. To distinguish a familiar regularity from a novel one, the specific tone-pip permutation must be remembered (we confirm this explicitly in Experiment 1B). Third, the task does not require listeners to memorize sounds *explicitly*: the emergence of the regularity readily pops out perceptually (see stimulus examples in supplementary materials). The task thus taps the process by which we automatically glean acoustic information from an ongoing sound-stream. Lastly, the sporadic presentation of REGr prevents them from becoming apparent to the listener, thereby allowing us to focus on putative implicit processes which underlie memory formation.

Across the experiments presented here, we ask whether human listeners form implicit long-term memories of sparsely reoccurring regular patterns (yes), whether this memory is robust to interference (yes), and whether it can be formed through passive exposure (partially). Through a combination of behavioural manipulation and modelling, we also demonstrate the interplay between short (a few seconds) and long (over many minutes) integration in the process of long-term memory formation. Overall, the results highlight the remarkable attunement of the auditory system to exceedingly sparse repeating patterns within the unfolding acoustic environment.

## Results

Participants listened to RAN, RANREG, RANREGr, CONT and STEP sequences as illustrated in *Figure 1* and were instructed to monitor for transitions. For each participant, different regularities were designated as reoccurring patterns (REGr). Critically, the RAN portion of RANREGr trials remained novel. Stimuli were presented in blocks of approximately 8 min each. Within each block, each REGr reoccurred three times (about 5% of the trials within a block) and was flanked by many novel patterns (RAN and RANREG).

The reaction time (RT) to STEP was subtracted from the RT to RANREG and RANREGr to estimate a lower bound measure of the time required to detect the emergence of regularity. RT values reported below are all baselined RTs (the raw RTs from which the RT to the STEP condition was subtracted).

Compared with RTs to the emergence of novel regularities (RANREG), we expected progressively faster RTs as regularities reoccur across the experiment (RANREGr), indicating that their representations have become retrievable from memory. We assess overall memory formation of REGr based on RTs averaged over all three reoccurrences within each block. However, we focus on RTs in each intra-block presentation to assess persistence of memory effects across experimental manipulations.

### Experiment 1A: implicit long-lasting memory for three reoccurring patterns

*Figure 2A-D* plots the mean and individual results of the regularity detection task performed in three sessions: five blocks on day 1, one block after 24 hr ('24 hr') and one block after 7 weeks, ('7 w'). Participants were highly accurate in detecting regularities (*Figure 2A*): d' plateaued at near ceiling performance after the first block. No difference was observed between hit rates for RANREG and RANREGr [no main effect of condition: $F(1, 19) = .39$, $p = 0.539$, $\eta_p^2 = .02$; no main effect of block: $F(5, 90) = 0.46$, $p = 0.804$, $\eta_p^2 = .02$; no interaction between condition and block: $F(5, 90) = 1.10$, $p = 0.367$, $\eta_p^2 = .06$].

Despite the ceiling effects associated with pattern detection (mean hit rate = 97.3%), faster RTs in RANREGr than in RANREG ('RT advantage') were observed in all participants by the end of the first session (block 5; *Figure 2D*), indicating a clear implicit memory for the reoccurring patterns. A repeated measures ANOVA on RTs with condition (RANREG and RANREGr) and block as factors yielded a main effect of condition [$F(1, 18) = 34.09$, $p < 0.001$, $\eta_p^2 = .65$], main effect of block [$F(5, 90) = 9.24$, $p < 0.001$, $\eta_p^2 = .3$] and an interaction between condition and block [$F(5,90) = 6.88$, $p < 0.001$, $\eta_p^2 = .28$]. Specifically, in the first block of the first session, performance did not differ

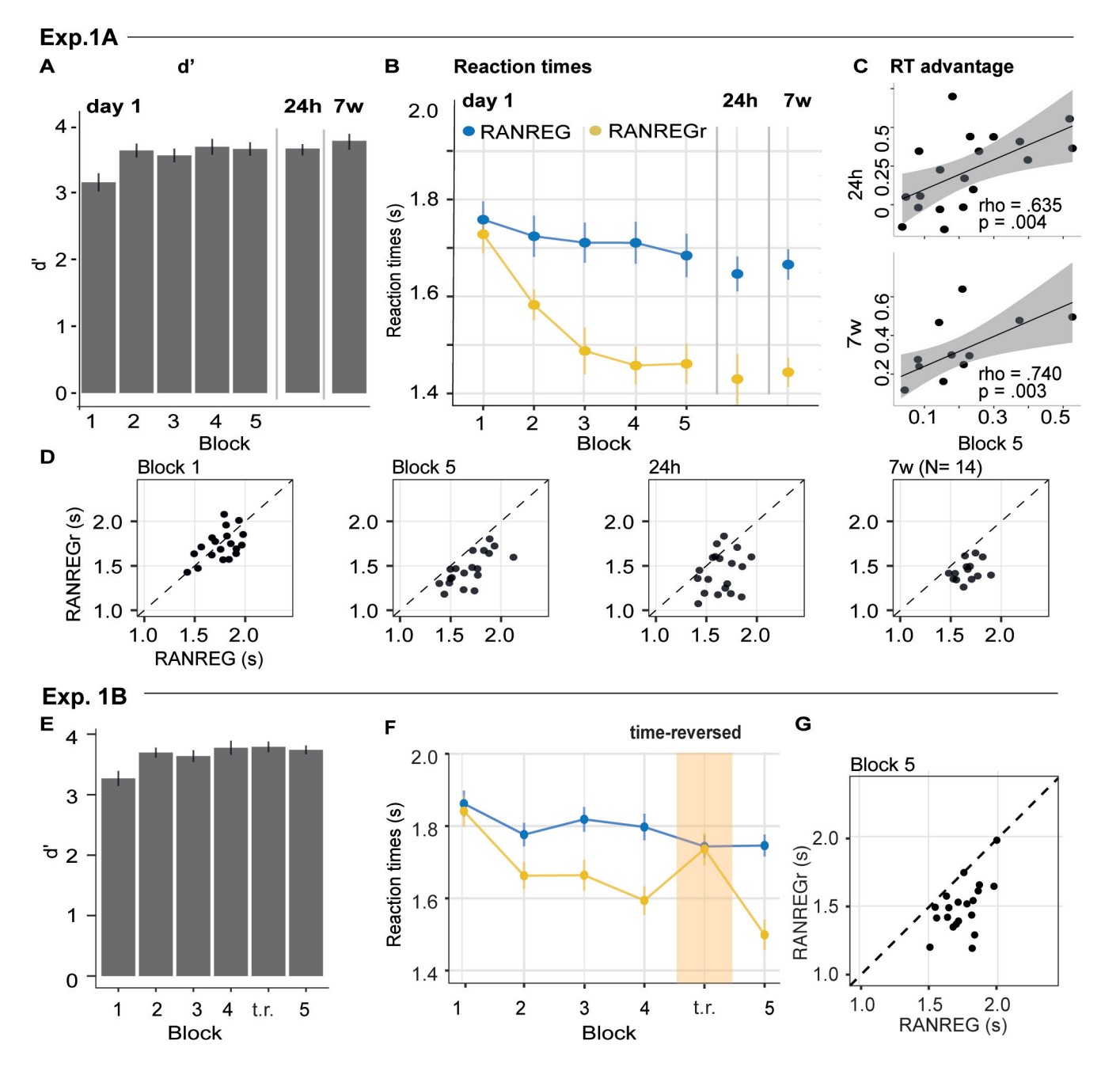

**Figure 2.** Experiment 1A (N = 19), 1B (N = 20): implicit long-lasting memory for three reoccurring patterns and specificity to sequential structure. (A–D) Exp. 1A (three reoccurring targets). (A) Sensitivity to emergence of regularity (d') across blocks during the first session, as well as after 24 hr and after 7 weeks. Error bars indicate 1 s.e.m. (B) RT to the transition from random to regular pattern in RANREG and RANREGr conditions, across blocks. Error bars indicate 1 s.e.m. '*Figure 2—figure supplement 1* plots the RT advantage for each intra-block presentation. (C) Correlations between RT advantage at the end of the first day – block 5 – and after 24 hr (upper plot) and after 7 weeks (lower plot). Each data point represents an individual. Note N = 14 in the 7W data due to attrition. (D) The relationship between RTs for the RANREG and RANREGr conditions. Each data point represents an individual participant. Dots below the diagonal reveal faster detection of RANREGr compared with RANREG. These implicit memory effects were not linked to explicit memory. See *Figure 2—figure supplement 2* for explicit recognition estimates. (E–G) Exp. 1B (time reversal): (E) Sensitivity to emergence of regularity (d') across blocks. (F) RT to the transition from random to regular pattern in RANREG and RANREGr conditions, across blocks. The block containing time-reversed REGr is shaded in yellow. The RT advantage dropped when REGr were time reversed, and restored in block 5. *Figure 2—figure supplement 3* plots the RT advantage for each intra-block presentation. (G) The relationship between RTs to the RANREG and RANREGr conditions in block 5.

*Figure 2 continued on next page*

*Figure 2 continued*

The online version of this article includes the following figure supplement(s) for figure 2:

**Figure supplement 1.** Experiment 1A. RT advantage for each intra-block presentation.
**Figure supplement 2.** Experiment 1A. Explicit recognition estimates.
**Figure supplement 3.** Experiment 1B. RT advantage for each intra-block presentation.

between RANREG and RANREGr [t(18) = 0.794, p = 1]. By the end of the second block (after 6 REGr reoccurrences), a significant difference (~140 ms; 2.8 tones) between RTs was observed [REG – REGr: t(18) = 3.964, p = 0.006]. This difference grew over the following blocks (all ps < 0.001), plateauing after block 3 (233 ± 0.17 ms; 4.7 tones). The RT advantage on the third block did not differ from the fourth [t(18) = −0.907, p = 1] nor from the fifth block [t(18) = −0.0003, p = 1]). In Experiment S1 (*Appendix 1—figure 1*), we demonstrate that similar effects are obtained when doubling the number of REGr patterns to be memorised (six different patterns per participant). In Experiments S2A and S2B (*Appendix 1—figure 2*), we further demonstrate that the memory trace is not abolished by introducing 'interrupting blocks' (in which REGr were not presented) between 'standard blocks' (in which REGr patterns reoccurred every ~3 min).

Critically, implicit memory for reoccurring regularities persisted after 24 hr and after 7 weeks: the RT difference between novel and reoccurring sequences remained constant between the last block of day 1 (block 5) and after 24 hr [t(18) = 0.139, p = 0.891], as well as between 24 hr and 7 weeks later [t(13) = −0.668, p = 0.515]. An inspection of intra-block reoccurrences (*Figure 2—figure supplement 1*) revealed that the RT advantage for REGr was similar between the third (last) intra-block presentation of day 1 and the first intra-block presentation after 24 hr [t(18) = 0.123, p = 0.903]; similarly, in the session conducted after 7 weeks, the RT advantage measured after the first intra-block presentation did not differ from the third (last) presentation in the session conducted after 24 hr [t(13) = 0 .958, p = 0.356; (*Figure 2—figure supplement 1*)]. This suggests that the effect observed after 24 hr and 7 weeks reflects the presence of a lasting memory trace of reoccurring regularities rather than rapid within-block re-learning.

Further, we examined the correlation of individual participants' RT advantage across the three sessions (*Figure 2C*). A robust correlation was found between the end of the first day (block 5) and the measurement taken after 24 hr (spearman's rho = 0.635, p = 0.004) – participants who exhibited a larger RT advantage at the end of the first day were also those showing a larger advantage 24 hr later. A similar correlation was found with performance after 7 weeks (spearman's rho = 0.740, p = 0.003). This confirms strong reliability of individual effects.

## The memory effects are not driven by explicit recognition of reoccurring patterns

Explicit memory for reoccurring regularities was examined at the end of each session by means of a familiarity task. Only regular sequences were presented: REGr (one presentation of each pattern) were intermixed with previously unheard REG patterns. Participants were instructed to indicate which patterns sounded 'familiar'. Classification was evaluated using the MCC score (see Materials and methods) which ranges between 1 (perfect classification) to −1 (total misclassification). Whilst low overall, the mean MCC on each testing session indicated above chance performance [day 1: mean = 0.231; t(18) = 4.214, p < 0.001; 24 hr: mean = 0.44, t(18) = 7.044, p < 0.001; 7 w: mean = 0.360, t(13) = 5.204, p < 0.001] (see *Figure 2—figure supplement 2*). An improvement in MCC scores was observed between day 1 and 24 hr later [t(18) = −3.635, p = 0.004], suggesting potential consolidation. There was no change in MCC scores between the 24 hr session and 7 weeks later [t(13) = 0.348, p = 1].

Importantly, MCC scores did not correlate with the RT advantage: MCC on day 1 did not correlate with the RT advantage observed in block 5 (spearman's Rho = 0.307; p = 0.201; a similar result was also obtained when pooling across participants from Exp. 1A and Exp. S1 (which used 6 REGr patterns, see Appendix 1) (Spearman's Rho = 0.114; p = 0.493; N = 38). Though a weak correlation between RT advantage and MCC was measured after 24 hr (uncorrected; Spearman's Rho = 0.459, p=0.048, N = 19), it disappeared after 7 weeks (Spearman's Rho = −0.024, p = 0.934, N = 14).

Therefore, implicit memory for reoccurring patterns, observed in nearly all participants, is not linked to explicit awareness of reoccurrence.

## Experiment 1B: Implicit memory is specific to sequential structure

To confirm that the RT advantage effects are driven by memory of sequential structure, we tested whether implicit memory for reoccurring patterns is tolerant to time reversal of the originally learned patterns (*Figure 2E–G*). Participants performed the regularity detection task as in Exp. 1A over six experimental blocks. The first four were identical to those in Exp. 1A. In the fifth block, REGr sequences were replaced by time-reversed versions. In block 6, the original REGr were introduced again. Participants were naive to the experimental manipulation. It was expected that, if implicit memory is specific to the sequential structure of regularity, the RT advantage should disappear in the time-reversed block (see also *Kang et al., 2017*).

Blocks 1–4 revealed the same effects as in Exp. 1A (*Figure 2F*) [main effect of condition: $F(1, 19) = 71.96$, $p < 0.001$, $\eta_p^2 = .79$; main effect of block: $F(3, 5) = 9.90$, $p < 0.001$, $\eta_p^2 = .34$; interaction condition by block: $F(3, 57) = 5.67$, $p < 0.001$, $\eta_p^2 = .23$]. Specifically, in the first block RTs in the RAN-REGr condition were similar to those in RANREG [$t(19) = 0.725$, $p = 1$], but became progressively faster (114 ms; 2.27 tones) in the second block [$t(19) = 3.56$, $p = .01$], and across the remaining blocks (all ps < 0.001) (203 ms; 4.1 tones in the 4th block).

Importantly, this RT advantage was abolished in the time-reversed block, but restored in the subsequent block containing the originally learned REGr: a repeated measures ANOVA with condition (RANREG and RANREGr) and the last two blocks as factors yielded a main effect of condition (F (1, 19) = 25.57, $p < 0.001$, $\eta_p^2 = .57$), a main effect of block (F(1, 19) = 18.09, $p < 0.001$, $\eta_p^2 = .49$), and an interaction condition by block (F(1, 19) = 40.03, $p < 0.001$, $\eta_p^2 = .68$), demonstrating the significantly greater RT advantage (RANREG novel – RANREGr) in the last than in the time-reversed block [$t(19) = 6.33$, $p < 0.001$]. The RT advantage for REGr in the third intra-block presentation of block 4 (*Figure 2—figure supplement 3*) was greater than in the first intra-block presentation of the time-reversed block [$t(19) = -2.261$, $p = 0.035$], but similar to the first intra-block presentation of the last block reintroducing the original REGr [$t(19) = 0.788$, $p = 0.440$].

These results constrain the nature of the observed memory effect to sequential information.

## Experiment 2: Limited formation of memory traces of non-adjacent patterns

We tested whether *adjacent* repetition of patterns (as is inherently the case for REG sequences) is required for implicit memory to be formed (*Figure 3*).

Over four blocks, listeners were exposed to RAN, RANREG and RANREGr trials as in previous experiments. We also introduced a new condition, PATinRAN (*Figure 3A*), which consisted of two identical *non-adjacent* 20-tone patterns (PAT) embedded within a random sequence of tone-pips. The second appearance always occurred at the end of the sequence. The first appearance was embedded partway through the sequence at an average distance of 1.7 s (range 0.5–2.9 s). To understand whether memories of non-adjacent patterns (PAT) can be formed during listening, three different PAT reoccurred three times within block (PATinRANr; the random parts of the sequences as well as the separation between the two PAT patterns remained random on each trial).

Both non-adjacent (PATinRAN, PATinRANr) and adjacent (RANREG, RANREGr) trials included two repetitions of each pattern with the only difference being that they were contiguous in the latter and separated by random tones in the former. Participants were instructed to respond if they detected two identical, not necessarily contiguous, 20-tone patterns within a trial; 50% of the trials consisted of fully random patterns. In order to make sure that participants paid equal attention to the (harder) PATinRAN sequences, accuracy was emphasized over response speed.

In the last block (block 5; 'test' block), we tested whether, following a comparable amount of exposure through block 1 to 4, PATinRANr and RANREGr patterns were similarly remembered. To equate difficulty of pattern detection in this block, PATinRANr sequences were replaced by versions where the two cycles were set adjacent. We refer to these conditions as RANREGr*. Participants were instructed to respond as quickly as possible. We compared the magnitude of the RT advantage associated with RANREGr* to that associated with RANREGr.

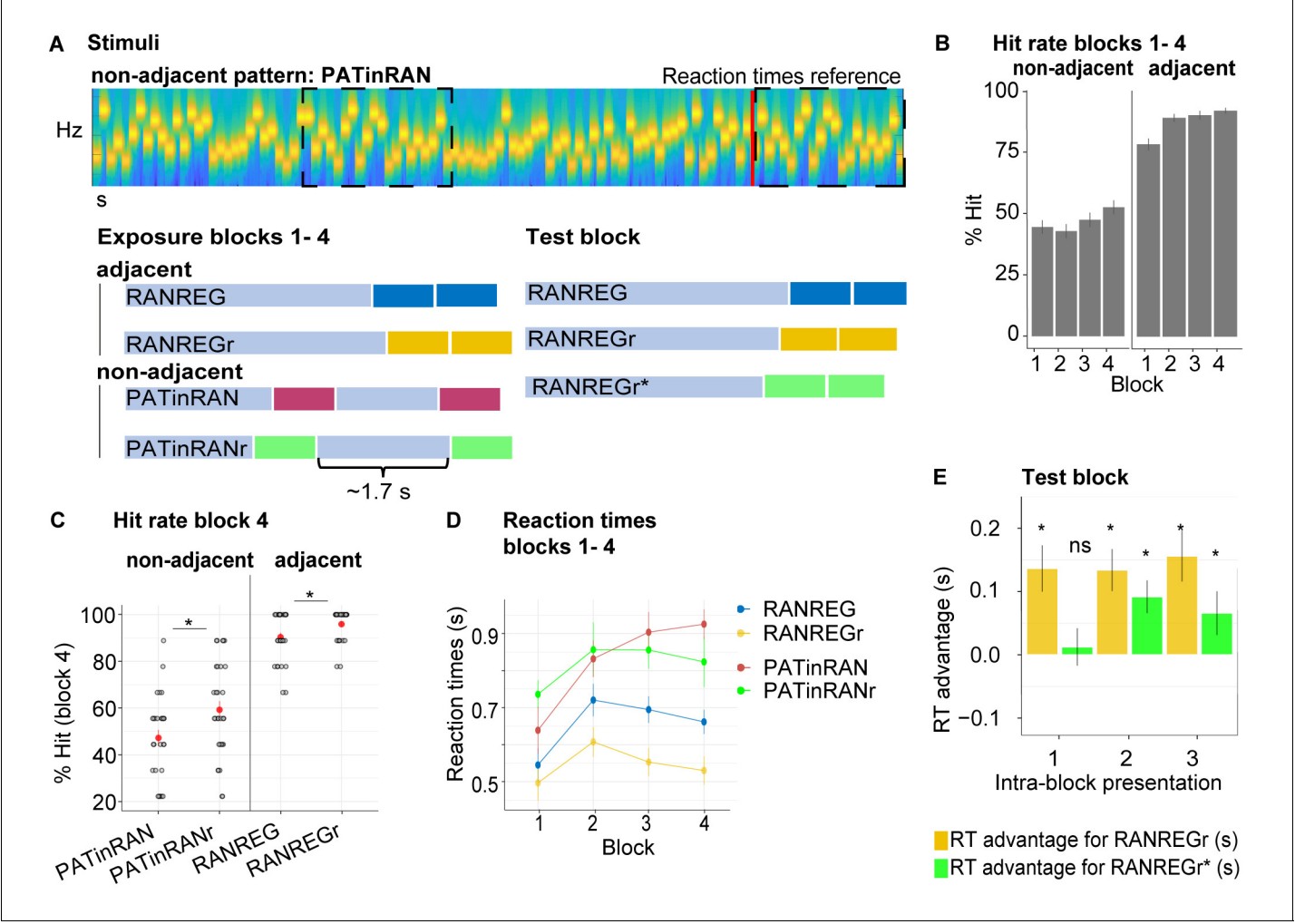

**Figure 3.** Experiment 2 (N = 30): Limited formation of memory traces of non-adjacent patterns. (A) In blocks 1 to 4, listeners were exposed to RAN, RANREG, RANREGr, PATinRAN and PATinRANr trials. An example spectrogram for a PATinRAN stimulus is provided. The non-adjacent repetitions of the 20-tones pattern (PAT) are indicated by dashed rectangles. In block 5 ('test' block) PATinRANr sequences were replaced by versions where the two cycles were set adjacent at the end of the trial (RANREGr*). (B) Accuracy (block 1 to 4): hit rates are computed separately for adjacent (RANREG and RANREGr) and non-adjacent (PATinRAN and PATinRANr) trials. (C) Hit rates in block 4, separately for novel and reoccurring adjacent and non-adjacent conditions. '*' indicates a significant difference between conditions. (D) RT (measured relative to the onset of the second cycle; see red line in A) across blocks 1 to 4 for RANREG, RANREGr, PATinRAN and PATinRANr. Error bars indicate 1 s.e.m. Note that since RT here is computed relative to the onset of the REG repetition, to compare RANREG RT with those reported in figures aboveadd 1 s. (E) Test block: RT advantage for RANREGr (yellow) and RANREGr* (green) in each intra-block presentation. Error bars indicate 1 s.e.m. To determine the presence of a memory trace to REGr* we specifically focus on the first intra-block presentation. '*' indicates a significant RT advantage, 'ns' indicates an RT advantage not significantly different from 0.

*Figure 3B* shows the detection performance during the exposure blocks (1 to 4). Despite having practised the PATinRAN condition, detection performance was overall worse, and substantially more variable in PATinRAN (mean over blocks 1–4: 47.36 ± 16.5%) relative to RANREG (88.47 ± 11.6%), and improved less across blocks [main effect of condition: $F_{(1, 29)}$ = 419.01, p < 0.001, $\eta_p^2$ = .94; main effect of block: $F_{(3, 87)}$ = 9.24, p < 0.001, $\eta_p^2$ = .24; interaction of condition per block: $F_{(3, 87)}$ = 4.83, p = 0.004, $\eta_p^2$ = .14]. Thus, whilst a pattern is highly detectable when contiguously repeated, performance drops substantially when the repetition is not adjacent, presumably due to limits on short-term memory.

Focusing on the 4th block (*Figure 3C*): a repeated measures ANOVA with the factors reoccurrence (novel/reoccurring patterns) and adjacency (adjacent/non-adjacent patterns) yielded a significant main effect of adjacency [$F_{(1, 29)}$ = 205.99, p < 0.001, $\eta_p^2$ = .88]. As expected, whilst participants were very apt at detecting RANREG patterns, performance on PATinRAN was

substantially more variable and lower overall. Interestingly a main effect of reoccurrence [F(1, 29) = 21.74, p < 0.001, $\eta_p^2$ = .43], was also observed, with no interaction between the two factors [F(1, 29) = 3.95, p = 0.056, $\eta_p^2$ = .12]. Therefore, detection data showed an increase in accuracy for reoccurring patterns in both adjacent and non-adjacent conditions. The emergence of this effect for RANREGr, despite its absence in Exp. 1A, is presumably driven by the below ceiling performance observed here (mean hit rate = 93% relative to 97.5% in Exp. 1A) – likely a consequence of the extra behavioural strain introduced by the PATinRAN stimuli. Critically, the finding of increased hit rates for PATinRANr (a mean increase of 15%) demonstrates that, through repeated exposure, listeners formed a memory trace for the non-adjacent patterns.

RT results across block 1 to 4 are shown in *Figure 3D*. To allow for a comparison across conditions, RTs here are measured relative to the onset of the second regularity cycle (indicated with a red line in *Figure 3A*). Since participants were encouraged to prioritise accuracy over speed in these blocks, the RT data in blocks 1–4 were not statistically analysed. However, an RT advantage (reaching 131 ms, 2.63 tones in block 4) is clearly visible for RANREGr relative to RANREG stimuli.

Test block: as a critical test for the formation of memory traces, we assessed the presence of an RT advantage in the 1st intra-block presentation of RANREGr and RANREGr* (*Figure 3E*). The RT advantage was significantly different from zero in RANREGr [one-sample t-test: t(29) = 3.724, p = 0.001], but not in the RANREGr* condition [one-sample t-test: t(29) = .419, p = 0.678]. A paired t-test further confirmed a greater RT advantage in the RANREGr than in the RANREGr* condition [t(29) = 3.169, p = 0.003]. This indicates that, as a group, participants did not demonstrate an immediate RT advantage to RANREGr* patterns. As seen in *Figure 3E*, an RT advantage in RANREGr* emerged following the second intra-block presentation. This effect may be associated with learning within the test block. A repeated measures ANOVA on RT advantage in the test block with the factors condition (REGr / REGr*) and intra-block presentation (1st / 2nd / 3rd) revealed a main effect of condition [F(1, 29) = 9.09, p = 0.005, $\eta_p^2$ = .24] but no main effect of intra-block presentation [F(2, 58) = 0.67, p = 0.515, $\eta_p^2$ = .02], or interaction [F(2, 58) = 1.27, p = 0.287, $\eta_p^2$ = .04], consistent with an overall smaller RT advantage to RANREGr*.

As an exploratory analysis, we tested whether higher detection accuracy for non-adjacent patterns (hit rates for PATinRANr / PATinRAN in block four) predicted a greater RT advantage when the patterns were set adjacently in the test block (REGr*). We observed a significant moderate correlation between the detection accuracy of PATinRANr in block four and the RT advantage in the 1st intra-block presentation of REGr* (spearman's rho = 0.429, p = 0.018) such that those participants who exhibited a higher detection accuracy for PATinRANr in block 4, also demonstrated a higher RT advantage for REGr* in the test block. This correlation with RT advantage was specific to PATinRANr, in that it did not extend to PATinRAN (spearman's rho = 0.017 p = 0.927) and held when the effect of detection accuracy for PATinRAN was accounted for (spearman's rho = 0.465, p = 0.011). The specificity to PATinRANr suggests that the link is not simply related to some property of short-term memory (in which case we would have expected a correlation with PATinRAN as well), but it is specific to the memory advantage for PATinRANr stimuli which developed over the first four blocks.

Overall, these results suggest the presence of measurable (though small) memory traces for reoccurring, non-adjacent patterns (PATinRANr). However, it is clear that the formation of robust implicit memory traces for sound sequences depends on short-term memory (and hence benefits from immediate repetition of patterns) such that introducing a gap of even 2 s results in substantially weakened storage in memory.

## Modelling

We constructed a 'memory constrained' computational model, based on 'prediction by partial matching' (PPM; see Materials and methods) to provide a formal simulation of the psychological mechanisms underlying the process of memory trace formation, as observed in Experiments 1A (*Figure 2*), 2 (Figure 3) and S2A (*Appendix 1—figure 2K*). These experiments reflect critical manipulations of the effect of long- and short-term memory decay. Although the existence of memory decay in humans is in general well established, ways of incorporating memory decay into probabilistic computational models of sequences processing is very much an active topic of research. Our PPM model implemented a single set of values (*Table 1*) that fully accounted for the dynamics of memory formation observed across experiments. As a benchmark, we also report the results for an equivalent

**Table 1.** Parameters for the memory-decay PPM model as manually optimized for Experiments 1A, 2, and S2A.

| Parameter | Value |
| --- | --- |
| Buffer capacity | 15 items |
| Buffer weight | 1 |
| Short-term memory weight[*] | 1 |
| Short-term memory duration[*] | 15 s |
| Long-term memory weight[*] | 0.02 |
| Long-term memory half life | 500 s |
| Long-term memory asymptote | 0 |
| Noise | 1.3 |
| Order bound | 4 |

[*]The combination of STM weight, STM duration and LTM weight yields a STM half-life of 3.06 s.

unconstrained model (i.e. with perfect memory), as employed in previous research using the same paradigm (*Barascud et al., 2016*).

The following cognitive hypotheses were instantiated:

1. Listeners learn sequence transition probabilities throughout the experiment. This approach is similar to other models of statistical learning (*Bröker et al., 2018*; *Harrison et al., 2011*; *Meyniel et al., 2016*; *Takahasi et al., 2010*) except the present model extends beyond first-order transition probabilities. Learning of sequence statistics is accomplished through partitioning the unfolding stimulus into sub-sequences of increasing order (n-grams) that are thereon stored in memory, such that the more a listener is exposed to a given *n*-gram, the stronger its salience ('weight'). Here, we allow *n* to range between 1 and 5, corresponding to Markovian transition probabilities of orders 0 to 4.

2. The listener uses these *n*-gram statistics to quantify the predictability (IC, where high IC corresponds to low probability and low IC corresponds to high probability) of incoming tones based on the preceding portion of the sequence and other information stored in memory as a generative probabilistic model (represented by PPM, see Materials and methods).

3. Sudden changes in IC are indicative of potential changes in the environment. In the present case, a sudden drop in IC reflects the onset of repetitive structure in the stimulus corresponding to a transition from RAN to REG. Once the model is sufficiently confident that a reliable drop has occurred, it registers a 'change detected' response analogous to the participant's button press.

4. The memory weight of a given n-gram observation decays over time, with this decay profile reflecting the dynamics of human auditory memory. In particular, we adopt the memory-weighting scheme recently presented in *Harrison et al., 2020*, and implement the following decay profile for the memory salience of an n-gram observation: a) an initial short and high-fidelity steady-state phase, representing an echoic memory buffer; b) a fast exponential-decay phase, representing short-term memory; c) a slow exponential-decay phase, representing longer-term memory (see *Figure 4A*, *Table 1* for more details). The model also adds noise to the memory retrieval stage, simulating inaccuracies in human memory retrieval.

Overall, the memory constrained model shows close qualitative correspondence to the pattern of RTs observed in Experiments 1 and 2, and specifically to the dynamics of the emergence of the RT advantage.

*Figure 5A* shows model outputs for experiment 1A using an unconstrained (left) and constrained (right) PPM model. The imposed memory constraints are able to reproduce the slow dynamics of REGr memory formation: like the human participants, the constrained PPM model experiences a moderate facilitation effect that grows over successive presentations of identical regular patterns. *Figure 4B* illustrates this effect in more detail, plotting average information content profiles for RANREGr trials in block five as compared to RANREGr trials in block 1.

It is important to note that the steady long-term decay, which is a key feature of the memory constrained model predicts that the performance facilitation should disappear after 24 hr, and certainly

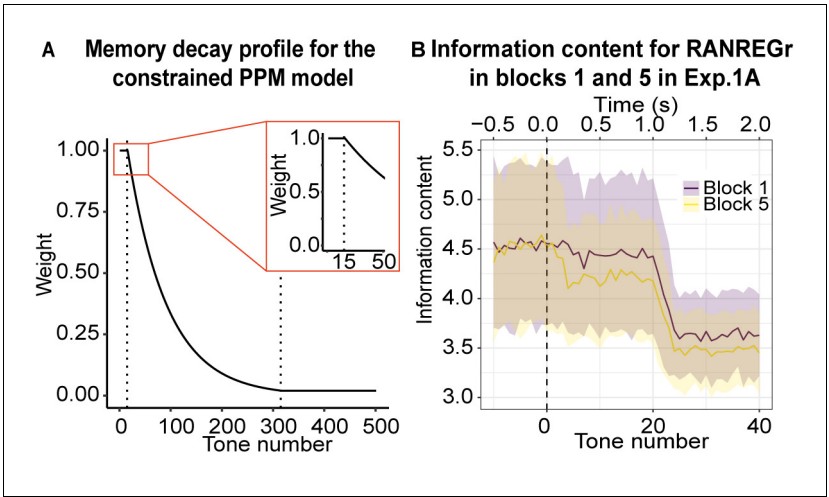

**Figure 4.** Memory constrained PPM model. (**A**) Memory decay profile for the constrained PPM model. The curve describes the weight of a given n-gram observation in memory as a function of the number of consequent tones that have been presented, assuming a constant presentation rate of 20 Hz. The two dotted lines indicate transitions between the different phases of memory decay: the first, between the memory buffer and short-term memory, and the second, between short-term memory and long-term memory. The inset shows the transition from the memory buffer (of 15 tones capacity) to the fast exponential-decay phase. See *Table 1* for model parameters. (**B**) Information content as a function of tone number for RANREGr trials in blocks 1 and 5 of Exp. 1A. Mean Information content is computed from the memory-decay PPM model, expressed in bits, and averaged over all trials. The shaded ribbons correspond to 1 STDEV. Trials are aligned such that a tone number of 0 corresponds to the first REG tone after the transition. The transition between RAN and REG phases becomes clearest after about 24 tones; however, the model detects the transition faster in block five than in block 1, because it partially recognises the REGr cycle from its previous occurrences, yielding a lower information content that is more clearly distinguishable from the RAN baseline and therefore requires less evidence accumulation time (=faster detection). However, it is obvious from the large error bars that the effects are subtle.

after 7 weeks. After such time periods, the memory traces for the reoccurring patterns should decay to zero, and the corresponding facilitation effect should disappear. Remarkably, the participants exhibited unaltered performance facilitation. This suggests that the memory traces of these reoccurring patterns are somehow 'fixed' at a certain point during testing. One way of simulating this effect would be to change the asymptote of the exponential memory decay, such that the memory trace asymptotically approaches a small but non-zero value as time tends to infinity. However, we found that incorporating such an asymptote caused the performance facilitation for RANREGr trials to increase constantly from block to block, in contrast to the slow plateau shown in the behavioural data. It seems likely, therefore, that there remains a non-trivial 'fixing' effect that may reflect consolidation processes, not accounted for by the current model (to our knowledge there is no other statistical learning model that accounts both for learning dynamics and long-term fixed effects).

Experiment 2 investigated the effect of pattern adjacency on pattern detection and memory formation. We trained unconstrained and constrained models on blocks 1–4, and report their performance for the 'test' block (block 5). As expected, the unconstrained PPM model is unaffected by adjacency (*Figure 5B* left). The memory-decay PPM model (*Figure 5B* right) fully reproduces the behavioural data (*Figure 3E*).

Overall, the modelling successfully replicated the slow dynamics of memory formation exhibited by human listeners demonstrating that memory constrained transition-probability learning is a plausible computational underpinning of sequential pattern acquisition.

## Experiment 3: Memories of a set of reoccurring regularities are not overwritten by subsequent memorization of another set

Does memorization of a new set of REGr interfere with the representation of a previously memorized set? Participants performed the same transition detection task as in Exp. 1A. They were exposed to a set of three reoccurring patterns (REGr1) in the first three blocks, followed by three blocks in which

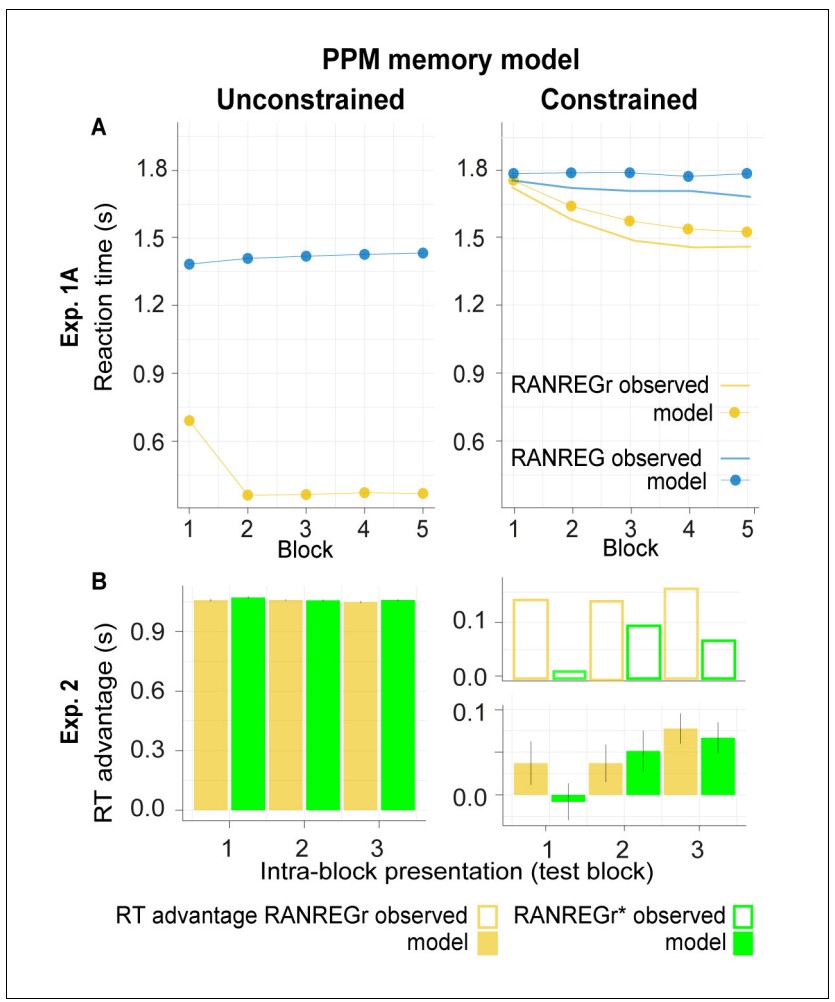

**Figure 5.** Model simulations for Experiments 1A and 2 for the unconstrained (left) vs. constrained (right) PPM model. Overall, we demonstrate a qualitative similarity between the formal simulation of constrained memory and observed human responses. (**A**) Exp. 1A: the estimated RTs to the transition from random to regular patterns in RANREG and RANREGr conditions across five consecutive blocks. For RANREG trials, the REG patterns are novel for each trial and the unconstrained PPM model detects transitions after one complete cycle plus eight tones (about 1.4 s; Note that the model change point detection algorithm was configured with a strict threshold in order to achieve an appropriate Type I error rate , see Materials and methods). For RANREGr trials after the first block, the regular patterns are already familiar from previous trials. The unconstrained PPM model remembers these previous patterns perfectly and hence demonstrates an immediate drop in RT. In contrast, the constrained model readily captures human performance, whereby the RT advantage for RANREGr trials slowly grows over successive presentations of the REGr patterns. (**B**) Exp. 2: RT advantage in RANREGr and RANREGr* conditions for each intra-block presentation within the test block. Data are presented in the same way as in *Figure 3E*. The unconstrained model reveals an equal RT advantage in both conditions. In contrast, as exhibited by the human listeners, the constrained memory model does not learn the reoccurring non-adjacent patterns across blocks 1 to 4, as shown by the null RT advantage in the first intra-block presentation in the RANREGr* condition. Error bars indicate 1 s.e.m.

another set of patterns (REGr2) reoccurred. Blocks 7 and 8 then re-tested memory for the reoccurring regularities of set 1 and set 2, respectively. After 24 hr, memory for the two sets was tested again.

Clear implicit memory for the first set of targets (REGr1), as indicated by an RT advantage, was observed after the 3rd block (*Figure 6B*) [main effect of condition: $F_{(1, 28)} = 41.01$, $p < 0.001$, $\eta_p^2 = .59$; main effect of block: $F_{(3, 84)} = 15.69$, $p < 0.001$, $\eta_p^2 = .36$; condition by block interaction: $F_{(3, 84)} = 6.83$, $p < 0.001$, $\eta_p^2 = .20$]. As expected, after three blocks of exposure the RT advantage in

the RANREGr1 condition (163 ms – 3.3 tones) was similar to that observed in Exp. 1A above. Critically, this RT advantage for RANREGr1 was not perturbed after the presentation of the second set of regularities (REGr2) [RT advantage in block three vs. block 7: t(28) = .877, p = 0.387]. It also lasted after 24 hr [RT advantage in block seven vs. after 24 hr: t(28) = −0.553, p = 0.584], and was similar to the 24 hr RT advantage observed in Exp. 1A [no main effect of experiment: F(1, 50) = .33, p = 0.567, $\eta_p^2$ = .01]. These results indicate that, once formed, memory traces are neither overwritten nor weakened by 'interfering' new sets of reoccurring patterns.

In blocks 4–6 presenting the second set of reoccurring regularities (REGr2) also showed an RT advantage, as demonstrated by the emerging separation between the RT to novel and reoccurring regularities. A repeated measures ANOVA on the RT advantage with 'experimental stage' (blocks 1–3, blocks 4–6) and block number (1st, 2nd or 3rd) showed a main effect of block number [F(2, 56) = 20.13, p < 0.001, $\eta_p^2$ = 0.42; consistent with a growing RT advantage across blocks], and stage [F(1, 28) = 15.70, p < 0.001, $\eta_p^2$ = 0.36] with no interactions. The main effect of stage suggests an overall larger RT advantage for the first set (REGr1). The noisier RT pattern observed in blocks 4–6 may be indicative of an order / fatigue effect. Importantly, at the end of day 1 the RT advantage for the two sets of reoccurring regularities did not differ (block seven vs. block 8: t(28) = 1.721, p = 0.096]. The RT advantage for the second set was maintained when tested after 24 hr (RT advantage of last block of day one vs. after 24 hr: t(28) = −0.277, p = 0.784), and did not differ from that of the first set [RT advantage after 24 hr for RANREGr1 vs. RANREGr2 t(28) = 1.848, p = 0.075].

## Experiment 4: Implicit memory is robust to pattern phase shifts

In all the previous experiments reoccurring regularities were always presented at the same phase of the REG cycle. Here we asked whether the resulting memory trace was anchored to this fixed boundary – that is, whether listeners remembered the pattern as a specific 'chunk' (*Dehaene et al., 2015*; *Thiessen, 2017*). If so, the RT advantage should reduce when REGr are phase shifted.

Listeners were presented with six reoccurring regularities (REGr) over three blocks. In block 4, identical REGr were presented but each presentation was with a shifted onset relative to the originally presented pattern (see *Figure 7A*, and Materials and methods).

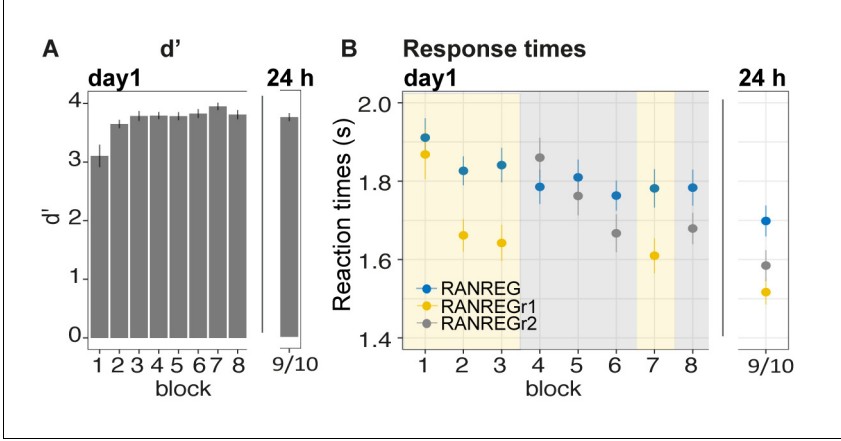

**Figure 6.** Experiment 3 (N = 29): memories of a set of reoccurring regularities are not overwritten by subsequent memorization of another set. Participants were exposed to a set of three reoccurring patterns in the first three blocks (REGr1, yellow shading), followed by three blocks in which another set of patterns was reoccurring (REGr2, grey shading). The final blocks (7 and 8) tested memory for set 1 and 2, respectively. After 24 hr, memory for the two sets was tested again. (**A**) d' across all blocks on day 1 and after 24 hr. Error bars indicate 1 s.e.m. (**B**) RT to the transition from random to regular pattern across blocks for RANREG, RANREGr1 and RANREGr2 on day 1 and after 24 hr. Error bars indicate 1 s.e.m. *Figure 6—figure supplement 1A* plots the RT advantage for each intra-block presentation. *Figure 6—figure supplement 1B* shows the RT data with N = 19.

The online version of this article includes the following figure supplement(s) for figure 6:

**Figure supplement 1.** Experiment 3.

*Figure 7C* shows the progressive emergence of the RT advantage associated with the memorization of the reoccurring patterns [main effect of condition: F(1, 19) = 21.12, p < 0.001, $\eta_p^2$ = .53; main effect of block: F(3, 57) = 18.52, p < 0.001, $\eta_p^2$ = .49; condition by block interaction: F(3, 57) = 10.64, p < 0.001, $\eta_p^2$ = .36]. Specifically, whilst in the first block performance did not differ between RAN-REG and RANREGr [t(19) = −0.876, p = 1], a faster RT to the RANREGr condition developed across ensuing blocks. This effect continued into block 4, where phase-shifting was introduced (*Figure 7C* bottom plot). The RT advantage for phase-shifted RANREGr (167 ms − 3.35 tones) in block 4 was greater than the RT advantage in block 3 (100 ms; 2 tones) [block three vs. block 4: t(19) = −13.111, p < 0.001] in the majority of participants (*Figure 7D*), demonstrating a strengthening (rather than disappearing) memory effect. The immediate robustness to phase shifting was confirmed by comparing the RT advantage in the first intra-block presentation in block 4, to that in the third (last) intra-block presentation in block 3 (*Figure 7—figure supplement 1*). No significant difference was observed [t(19) = 1.069, p = 0.298], supporting the conclusion that the RT advantage persisted despite phase shifting.

Further tests confirmed that the RT advantage for REGr in block 4 was similar across small and large phase shifts: a repeated measures ANOVA with factor phase shift (small / large, namely 1–5

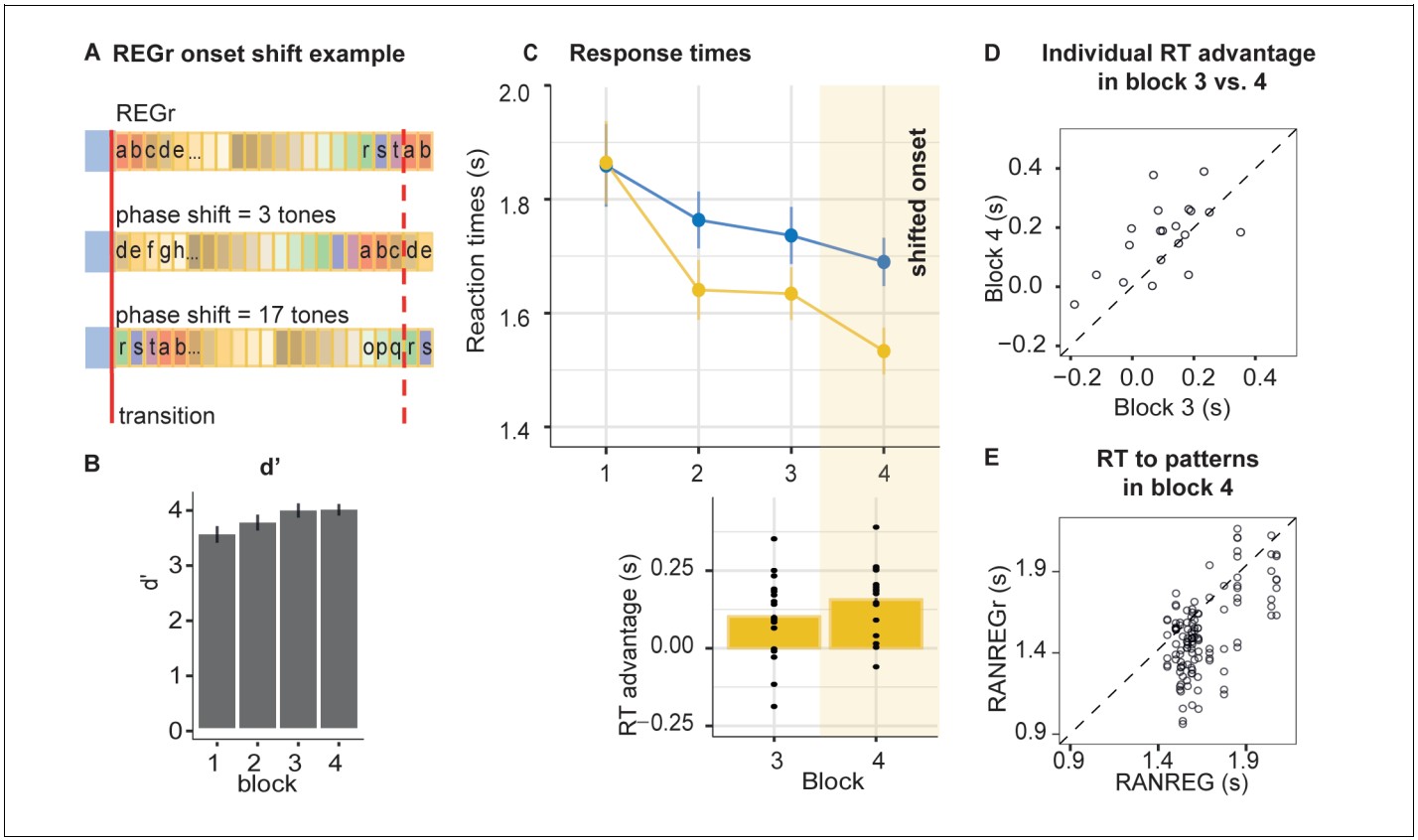

**Figure 7.** Experiment 4 (N = 20): Implicit memory is robust to pattern phase shifts. (**A**) In this experiment, six different reoccurring regularities (REGr) per participant were presented. In block 4 (yellow shading in C), these patterns were replaced by versions with shifted onset relative to the originally learned REGr. Two examples of phase shifted REGr and their original REGr version are depicted. The solid red line indicates the transition between RAN and REG (the onset of the regular pattern); the dashed red line denotes one cycle (20 tones) (**B**) d' across all blocks. Error bars indicate 1 s.e.m. (**C**) RT to the transition from RAN to REG pattern across blocks for RANREG and RANREGr. The bottom plot represents the RT advantage observed in blocks 3 and 4. Error bars indicate 1 s.e.m. *Figure 7—figure supplement 1* plots the RT advantage for each intra-block presentation. (**D**) The individual RT advantage in block three compared with block 4. Each circle represents an individual participant. (**E**) Plotted is the relationship between RTs to RANREG and RANREGr in block 4. Each circle represents a unique REGr pattern (six per participant), plotted against the mean RT to RANREG for that participant.

The online version of this article includes the following figure supplement(s) for figure 7:

**Figure supplement 1.** RT advantage for each intra-block presentation.

and 16–19 vs. 6–15 tones from the original onset) yielded no significant effect of phase shift on the RT advantage [F(1, 19) = 0.74, p = 0.400].

These results suggest that sequences are not represented as a fixed chunk of sequential items which is retrieved as a single unit, but more likely as a collection of sequential predictions that are flexibly retrieved from memory according to the available sensory information.

As a further probe into the nature of the representation of the pattern in memory, in Experiment S3 (*Appendix 1—figure 3*) we investigated listeners' tolerance to small frequency transpositions. We reveal a transfer of the RT advantage to the transposed pattern, suggesting that the formed representation is not of an exact echoic nature. It is possible that tolerance to frequency transposition reflects a 'fuzzy' spectral representation, although we note that the spacing in the present pool – 12% – is generally larger than the just noticeable difference (JND) for frequency typically exhibited by non-musically trained listeners (*Tervaniemi et al., 2005*). Alternatively, the tolerance to transposition may suggest that instead of the specific frequency pattern, the auditory system maintains a representation of the contour, or inter-tone interval within the pattern.

## Experiment 5: Implicit memory can form when sounds are behaviourally irrelevant, but does not immediately transfer to behaviour

We asked whether memories for reoccurring patterns are formed when sequences are not behaviourally relevant. Naïve participants were exposed to three blocks of the same kind as in Exp. 1A, but instructed to detect the STEP changes only, and ignore the other sounds. In the fourth block ('test' block), they were instructed to also detect the RANREG transitions.

We analysed the performance in the test block of the pre-exposed group in comparison to the performance of a non pre-exposed 'control' group, formed by pooling block one data from several other experiments (*Pooled data-block₁*, N = 147, see Materials and methods). Sensitivity to transitions in the test block (*Figure 8A*) was high overall (mean d' = 2.77 ± .73), but lower than in the first block of the control group [independent sample t(163) = −2.028, p = 0.044]. This is likely because, in order to keep them naive, participants did not receive training on RANREG detection.

In the test block (*Figure 8B*), the mean RT to RANREGr was significantly faster than that to novel RANREG [t(17) = 3.1, p = 0.006], consistent with the presence of an RT advantage. The RT

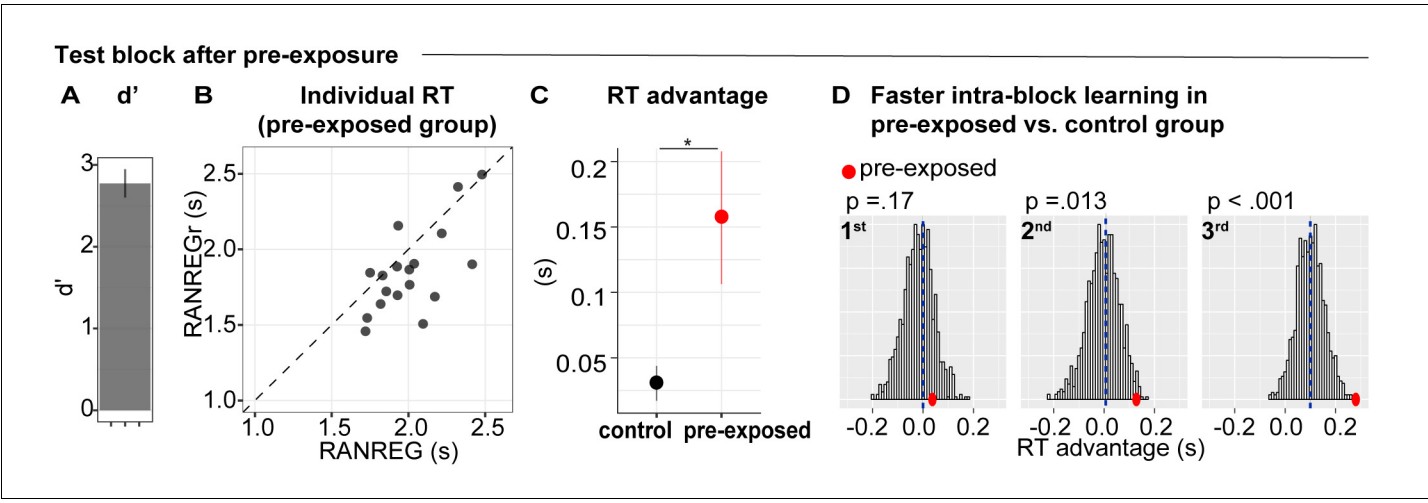

**Figure 8.** Experiment 5 (N = 18): implicit memory can form when sounds are behaviourally irrelevant, but does not immediately transfer to behaviour. During three initial blocks, participants were asked to respond only to the STEP trials and ignore the other sounds. In the following test block, they were instructed to also detect the RANREG transitions. (A) Sensitivity to emergence of regularity (d') in the test block. Error bars indicate 1 s.e.m. (B) The relationship between RTs to the RANREG and RANREGr conditions in the test block. Each data point represents an individual participant. Dots below the diagonal indicate faster detection of RANREGr compared with RANREG. (C) RT advantage in the pre-exposed and the control group (participants without previous exposure; see Materials and methods). Error bars indicate 1 s.e.m. '*' indicates a significant difference. (D) Bootstrap resampling-based distributions of the RT advantage for the 1st, 2nd and 3rd intra-block presentation from the control group. The mean of the distribution is indicated by blue dashed lines. Red dots indicate the data from the present experiment (pre-exposed group). One-tailed p-values are reported with each graph.

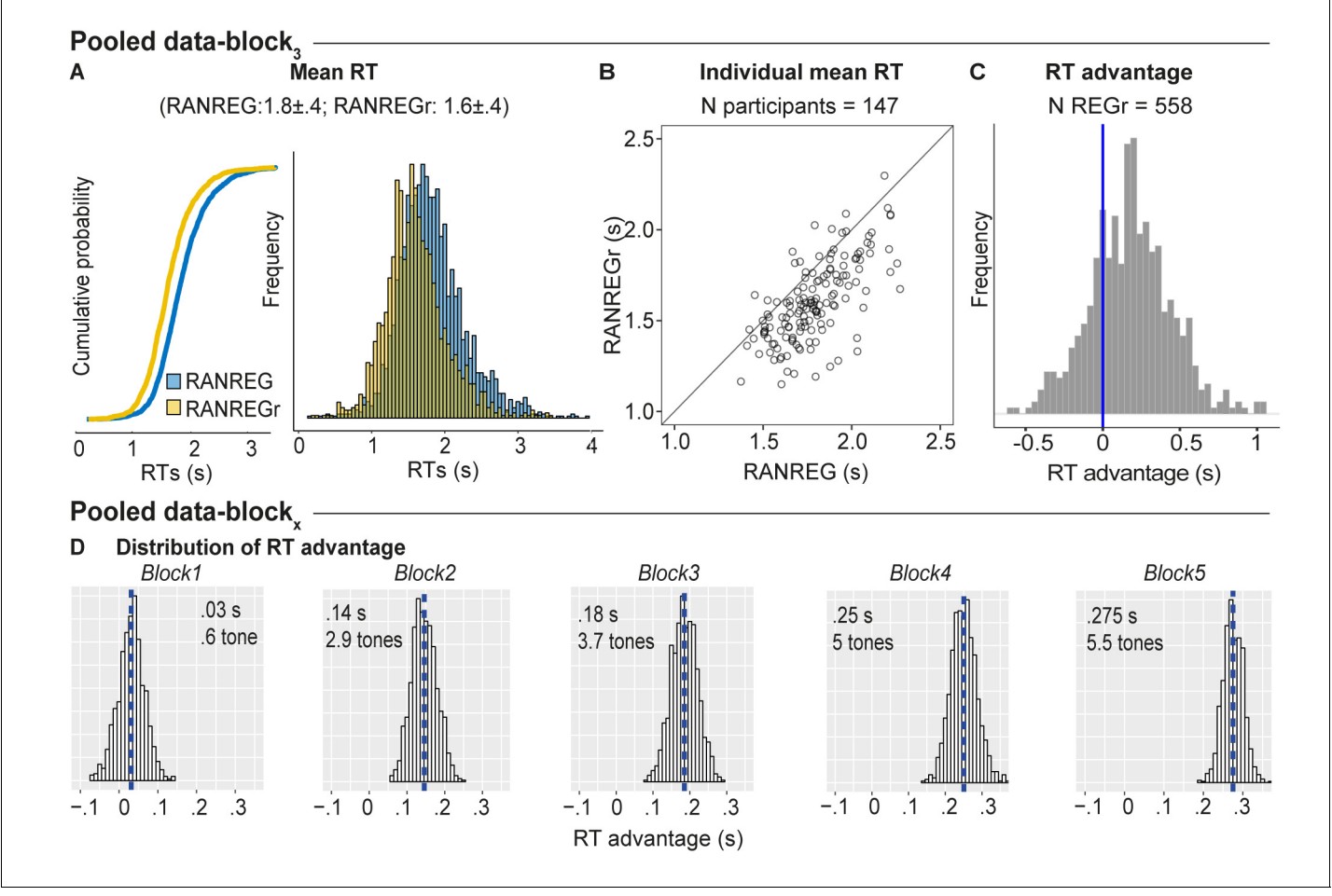

**Figure 9.** Individual variability in implicit memory. (**A**) Cumulative distribution function (left) and distribution (right) of RTs to RANREG and RANREGr pooled from block 3 of several experiments (see Materials and methods). A two sample Kolmogorov-Smirnov test confirmed a significant difference in cumulative probability (D = 0.232, p < 0.001) (**B**) The relationship between RTs to the RANREG and RANREGr conditions in block 3. Each circle represents an individual participant. (**C**) Distribution of RT advantages across 558 different REGr patterns as measured after three blocks (9 presentations of REGr). Values > 0 indicate faster RTs to REGr relative to novel REG. (**D**) Distributions of the RT advantage in each block. To estimate the distribution of the RT advantage across the population (of young, healthy participants) we pooled data from several experiments (see Materials and methods) in which participants performed the standard regularity detection task. Pooled data-block$_1$ reflects the distribution of RT advantage after one block (3 presentations of REGr), Pooled data-block$_2$ reflects the distribution of the RT advantage after two blocks (6 presentations of REGr), etc. The distributions are computed via a bootstrapping process whereby on each iteration (1000 overall), data from 20 participants are chosen randomly (with replacement), to obtain an average RT advantage. The mean of each distribution is indicated by blue dashed lines. Overall these distributions demonstrate a robust emergence of an RT advantage after the first block.

advantage in the pre-exposed group (~157 ms, 3.14 tones) was substantially greater than in the control group (~30 ms, 0.6 tones) [independent sample t(163) = 3.023, p = 0.003], indicating a beneficial effect of pre-exposure.

As a critical test for the presence of a memory trace after pre-exposure, we examined RT in each intra-block presentation of REGr. If memories for reoccurring patterns are formed during pre-exposure, an RT advantage should be exhibited immediately - at the first presentation of REGr in the test block. One sample t-tests demonstrated that an RT advantage was absent at the first and second intra-block presentations [t(16) = 0.377, p = 0.711; t(17) = 1.691, p = 0.109], but emerged at the third presentation of REGr [t(17) = 3.954, p = 0.001]. We also compared the RT advantage, across intra-block presentations, between the pre-exposed and control groups. A bootstrap approach (see Materials and methods) was used to generate a distribution of performance over subsets of 20 participants drawn from the control group and to compare with the actually observed performance in the pre-exposed group (*Figure 8D*). The plots in *Figure 8D* show distributions of the RT

advantage for the 1st, 2nd and 3rd REGr presentation in the control group. The mean RT advantage of the 'pre-exposed' group is shown by the red dots. This analysis revealed that the RT advantage to the 1st presentation did not differ from the control group. However, a difference emerged after the 2nd presentation. This suggests that, by the 2nd appearance of REGr in the 'test' block, the passively pre-exposed group exhibited substantially faster responses than non pre-exposed participants. The difference between the passively pre-exposed group and the control group grew further by the 3rd presentation.

Overall, these results demonstrate that implicit memory was not present at the onset of the test block (as evidenced by the lack of an RT advantage); however, learning occurred more rapidly in the pre-exposed listeners such that by the end of the test block, they exhibited a substantially higher RT advantage than that shown by the control group.

Explicit memory was poor (mean MCC = 0.064) and did not correlate with the RT advantage measured in the test block [Spearman's Rho = 0.235; p = 0.347].

### Across-experiment analysis reveals that most patterns are remembered and most participants exhibit implicit memory

We quantified the robustness of the memory effect for reoccurring patterns across the different experiments reported here. *Figure 9A* shows the distribution of RTs for RANREG vs. RANREGr pooled from block three data, (i.e. after nine presentations of each REGr; approx. 25 min of listening) where most data from different experiments were available (the pilot experiment, Experiment 1A, 1B, 3, 4, S1, and S3). In *Figure 9B* each dot represents the mean RT for RANREG vs. RANREGr of an individual participant (N = 147). 88.4% of participants exhibited an RT advantage, which we interpret as revealing implicit memory for REGr.

We also tested the generality, across patterns, of the observed memory effect. It is important to note that all REGr were similar in the sense that all are composed from the same set of tones and only differed in the specific permutation of their order. *Figure 9C* plots a distribution of the RT advantage per unique REGr (558 overall). Though the data are inherently noisy (RT is quantified as an average over only three presentations in block 3), RT advantage appears to be normally distributed with 75.6% of patterns exhibiting a memory effect. This demonstrates that the observed effects are not driven by particularly 'memorable' REGr sequences. The same analysis run over block five data (not shown; N unique REGr = 165) showed that 84.4% of REGr were associated with an RT advantage after 15 reoccurrences. *Figure 9D* plots the distributions of group RT advantage per block, based on performance observed across all of the experiments reported (see Materials and methods). A gradual build-up of RT advantage is seen across blocks reaching a mean of 5.5 tones by the end of block 5.

Overall the results demonstrate that the memory effect generalizes to most (healthy, young) listeners and is not driven by particular memorable stimuli.

## Discussion

We used rapid sequences of discrete sounds (*Barascud et al., 2016*; *Southwell et al., 2017*; *Zhao et al., 2019*) specifically structured to allow for detailed behavioural and model-based investigation of memory formation. All sequences were generated from a fixed set of 20 frequencies, with the only difference being the order in which these were presented. Participants performed a regularity detection task and were oblivious to rare reoccurrences of certain patterns throughout the session. However, reaction times to new vs. previously encountered regularities demonstrated that following limited exposure to reoccurring patterns listeners retained sequential information in long-term memory. Statistics of pattern learning across experiments revealed that most patterns were remembered, and most participants exhibited a memory effect, although the size of this effect varied across individuals. Memory was implicit, resistant to interference, and preserved over remarkably long durations (over 7 weeks). Importantly, we also demonstrate that local pattern repetition was critical for long-term memory formation. This finding highlights a key role for immediate reinforcement and implicates an interplay between rapid and slow memory decay in supporting the formation of enduring memories of arbitrary sound sequences.

Overall the results reveal the brain's remarkable capacity to implicitly preserve arbitrary sequential information in long-term memory.

## Relationship to 'noise memory'

The general behavioural pattern revealed here is reminiscent of the 'noise memory' effect first shown by *Agus et al., 2010* (see also *Agus and Pressnitzer, 2013*; *Andrillon et al., 2015*; *Gold et al., 2014*; *Keller and Sekuler, 2015*; *Luo et al., 2013*). In that study naïve listeners readily remembered reoccurring white-noise snippets presented amongst novel noise bursts. The learning was unsupervised, rapid, implicit and lasted upwards of 2 weeks.

Inspections of the nature of this memory revealed that it was robust to time reversal and even to scrambling into bins as small as 10–20 ms, indicating that the remembered features reflect local spectro-temporal idiosyncrasies within the reoccurring noise snippet (*Agus et al., 2010*; *Viswanathan et al., 2016*). The apparent dependence of this memory on certain local features of the noise signal may also explain the high inter-sample variability often seen with this paradigm (e.g., the distinction between 'memorable' and 'not memorable' patterns; *Agus et al., 2010*; *Viswanathan et al., 2016*; *Kang et al., 2017*).

In contrast, here we focus on fast memory formation for *sequences* of discrete tones, distinguishable only by their specific order, and presented in a surrounding context of highly similar patterns (all sequences consisted of the same 20 'building blocks'). We showed that the vast majority of patterns were learned, revealing high sensitivity to reoccurring arbitrary frequency patterns despite the exceedingly rare reoccurrence rate (every ~3 min; 5% of trials; in contrast to the much more frequent reoccurrence, <~15 s in *Agus et al., 2010* and *Kang et al., 2017*).

An important question for future work will be to determine whether these effects draw on similar or distinct neural systems (discussed further below).

## Memory for auditory sequences

Signals based on tone-pip patterns have long been used to understand the effect of auditory memory on listeners' perception of sound sequences (e.g. *Watson et al., 1975*; *Atienza and Cantero, 2001*; *Näätänen et al., 1993*; *Schröger et al., 1992*; *Tervaniemi et al., 2001*; *Moldwin et al., 2017*). However, these paradigms are predominantly based on extensive exposure (in the order of hundreds of consecutive repetitions) to a single pattern.

Of particular relevance is a large body of work, broadly referred to as 'statistical learning', which has demonstrated the brain's capacity to discover repeating structure in random stimulus sequences (*Conway and Christiansen, 2005*; *Frost et al., 2019*; *Kim et al., 2009*; *Saffran et al., 1999*; *Saffran and Kirkham, 2018*). The classic paradigm (*Saffran et al., 1996*; *Santolin and Saffran, 2018*) involves a small set of syllables arranged into short 'words' (e.g., three syllables each). A few minutes' exposure to such structured streams leads to learning of the statistical structure of the unfolding sequence such that subjects can distinguish the repeatedly occurring 'words' from a random arrangement of syllables.

Our results can be interpreted as reflecting similar implicit learning processes. However, in contrast to the demonstrations above which usually involved one or a small number of stimuli that are repeated many times, we show that a very sparse presentation of long patterns, which are intermixed with many highly similar sequences, is sufficient for robust memories to be formed.

Note that to focus on implicit memory formation, we placed our listeners in rather extreme conditions, both in terms of presentation rate of reoccurring targets and their complexity. It is possible that relaxing these constraints would result in stronger (but perhaps more explicit) memories.

We showed that listeners can learn at least six concurrently presented REGr patterns (Exp. 4 and Exp. S1 in Appendix 1). Important questions for future work involve understanding the capacity limits on this memory and the factors which might affect subsequent forgetting.

Overall, we demonstrate that the brain is tuned to retain repeated structure in our acoustic environments, even when such reoccurrences are exceedingly infrequent and the signals are highly similar.

Preserving as much information as possible from the unfolding sensory input is important for an organism because the relevance of any single event is not always immediately apparent, but is rather inferred post-hoc, e.g., through repetition ("I've heard this exact pattern twice within 3 min, therefore it might reflect an individual sound source rather than random noise in the forest"; e.g., *McDermott et al., 2011*; *Woods and McDermott, 2018*). Our results hint at the heuristics utilized by the brain in determining how representations of statistical structure in the sensory environment

are converted from transient to stable forms of memories (*Leimer et al., 2018*; *Li and van Rossum, 2020*).

## Reaction time as a measure of memory formation

We used reaction time (RT) as a proxy for memory formation. RT allowed us to determine how much information was required for listeners to detect repeating (REG) structure and to compare these measures with formal models of sequence processing. We hypothesized that reoccurrence would increase the weight of sequence components in memory resulting in faster detection of regularity. RT thus provided a sensitive means for tracking the formation and maintenance of such memories over time.

We observed that the RT to REGr steadily shortened with increasing number of reoccurrences, allowing us to measure the dynamics of memory trace establishment. The 'RT advantage', defined as the difference in RT between novel and reoccurring REG patterns, grew rapidly over the first three blocks (9 reoccurrences) and then stabilized, though evidence from *Figure 9D* suggests a continuous slow growth throughout the experimental session. The absence of correlation between the familiarity test and the RT advantage suggests a dissociation between implicit memory and explicit recall abilities.

## Time scales of memory for sequences

The basic behavioural task required participants to detect the transitions from RAN to REG – namely the emergence of repeating structure. As such it fundamentally relied on auditory short-term memory: in order to detect REG patterns, the listener must compare incoming tones to those that occurred at least a cycle ago. The effect of reoccurrence suggested that listeners also draw on much longer-term memory whereby information about previously encountered sounds is retained over minutes between successive REGr presentations.

Due to practical issues related to providing breaks, all of the reported experiments were subject to fixed presentation parameters: the experimental session was divided into blocks of roughly 8 min and REGr were presented three times per block. We therefore only have a relatively coarse estimate of the properties of the long-term memory store. Lengthening of inter-reoccurrence intervals was probed by introducing interrupting blocks where only novel patterns were presented (see Exp. S2A-B in Appendix 1). Memory was largely maintained over ~10 min intervals indicating a very slow long-term decay. In conjunction, the results of Exp. 2 suggested that the short-term memory store is critical for long-term memory formation. Participants were markedly impaired at detecting pattern repetition when the two cycles were separated by a brief series of random tones (about 2 s). Those conditions were also associated with substantially reduced long-term memories for the reoccurring patterns, indicating that immediate reinforcement is critical for the formation of lasting memory traces. These observations point to an integral interplay between a short (few seconds) and much longer (at least a few minutes) integration in the process of formation of robust, implicit memories for reoccurring arbitrary sound sequences.

The persistence of a stable RT advantage 24 hr and 7 weeks after initial exposure demonstrates the establishment of a long-term memory representation, possibly through a process of consolidation involving long-lasting synaptic changes (*Phan et al., 2017*; *Redondo and Morris, 2011*). It may also be tempting to interpret the resistance to interruption, observed in early stages of memory formation (Exp. 3, Exp. S2 in Appendix 1), as a hint that a form of consolidation might have occurred already after a few initial presentations.

In animal models, repetitive exposure to sound tokens (though, notably at a much higher rate than that used here) has been shown to evoke a process of long-lasting adaptation manifested as sparser firing and increased response specificity. These effects, persisting for hours to days after the initial stimulation, have been observed in primary and secondary auditory areas in song birds ( Caudal Medial Nidopallium; *Cazala et al., 2019*; *Honda and Okanoya, 1999*; *Lu and Vicario, 2014*; *Menyhart et al., 2015*; *Takahasi et al., 2010*; *Chew et al., 1996*; *Soyman and Vicario, 2019*) and in secondary auditory cortex in ferrets (*Lu et al., 2018*). The hypothesis that similar processes might back the behavioural effects we report is appealing.

Agus et al. proposed that mechanisms based on spike-timing-dependent plasticity (STDP; *Markram et al., 1997*; *Masquelier et al., 2008*; *Masquelier et al., 2009*) may be possible neural

underpinnings for rapid noise memory formation: repeatedly presented, but relatively temporally confined, spectro-temporal 'constellations' within the noise snippets may evoke coincident firing among auditory afferents leading to rapidly emerging selectivity for this feature in subsequent presentations of the same noise burst. *Kang et al., 2017* suggested that including a degree of temporal integration can also account for similar effects observed with temporal patterns (*Kang et al., 2017*; *Karmarkar and Buonomano, 2007*; *Lim et al., 2017*; see also *Bi and Poo, 2001*). As will be discussed below, the behavioural pattern observed here is consistent with sequential information being stored as short sub-sequences (n-grams), that is without retaining the full 20-item sequence. Therefore, a form of STDP, incorporating an integration time of several hundred milliseconds, may underpin the representation of n-grams and implement their increased weight through reoccurance, thus supporting memory for discrete tone sequences.

On a systems level, accumulating evidence suggests that an interaction between auditory cortex and the hippocampus may play a role in memory formation. Previous work has implicated the hippocampus in sensitivity to sensory patterns across rapid time scales (*Aly et al., 2013*; *Stachenfeld et al., 2017*; *Yonelinas, 2013*) and specifically in the process of discovering RAN-REG transitions (*Barascud et al., 2016*). There is also some evidence that hints at its possible role in supporting long-term memory for acoustic patterns (*Kumar et al., 2014*).

## What is being remembered?

The RT advantage to REGr reflects an implicit memory of sequential structure (Exp. 1B). But what, specifically, is remembered? Clearly participants did not perfectly memorize the full pattern, in that this would have been associated with much faster RTs (e.g. as exhibited by the observer with unconstrained memory, *Figure 5A*). Instead, by the end of block 3, the distribution of RT to REGr shifted leftwards by about four tones, without otherwise changing (*Figure 9A*). Modelling suggests that this performance is consistent with a statistical-learning effect whereby the participants retained imperfect memory of patterns presented earlier in the experiment. These memories are not strong enough to prompt immediate recognition of a pattern heard in a past trial, but they are sufficiently strong to speed the recognition of that pattern once it begins repeating in the new trial.

Similar to other models of statistical learning (*Bröker et al., 2018*; *Harrison et al., 2011*; *Meyniel et al., 2016*), our memory-constrained PPM model explicitly assumes that listeners represent the unfolding sequences in the form of n-gram sub-sequences of variable length, from which transition probabilities are computed. Previous computational, behavioural and neuroimaging studies *Bianco et al., 2020*; *Conklin and Witten, 1995*; *Di Liberto et al., 2020*; *Egermann et al., 2013*; *Pearce et al., 2010*; *Pearce and Wiggins, 2004*; *Pearce and Wiggins, 2006* demonstrated that PPM successfully generalizes to prediction of musical sequences and effectively accounts for psychophysiological responses to melodies. In particular, PPM provided a good match to brain response latencies evoked by transitions between RAN and REG patterns (*Barascud et al., 2016*; *Southwell and Chait, 2018*), suggesting that listeners may rely on similar memory representations as those proposed by the model. Here, the memory constrained version of PPM was able to successfully simulate human performance - concretizing how the interplay between short- and long-term decay might give rise to the progressive emergence of a memory trace across presentations. Whether listeners do indeed represent auditory patterns in this way is a matter of ongoing debate (e.g. *Thiessen, 2017*). Additional support for an n-gram-like representation is provided in Exp. 4, which demonstrated that the REGr RT advantage is robust to pattern phase shifts. This finding indicates that REG patterns are not encoded in memory as rigid chunks of sequential items (*Perruchet and Pacton, 2006*; *Thiessen, 2017*), but are instead represented as a transition rule which allows for flexible retrieval. Whilst further empirical evidence is essential to determine the nature of the memory representation, the insight into single-trial level dynamics derived from the present modelling (*Figure 4*) may be useful for constraining the search for the physiological underpinnings of these phenomena. Furthermore, the model can readily be applied to statistical learning in other modalities (reviewed by *Frost et al., 2019*) and even in other species, including songbirds such as finches, known to be capable of statistical learning (*Menyhart et al., 2015*; *Takahasi et al., 2010*).

A related question pertains to the generalizability of the present model to natural sounds beyond quantized sequences, such as those used here. In order to relate listeners' performance to a measure of statistical information within unfolding signals, simplifying assumptions are necessary. This

includes the presence of a prior stage of category formation which converts a continuous sound into discrete units that form the model's 'alphabet'. Accumulating evidence is indeed revealing that unsupervised segmentation of unfolding sounds into basic elements, perhaps using envelope-based cues, may be an inherent feature of listening (*Ding et al., 2017*; *Doelling et al., 2014*; *Hickok and Poeppel, 2007*; *Poeppel, 2003*).

### Does sequence memory require attention?

The short-term memory mechanisms which allow listeners to discover the emergence of repeating structure (RANREG) in rapid tone sequences have been demonstrated to operate automatically, even when listeners' attention is directed away from sound: brain activity recorded from naïve, distracted listeners reveals robust responses to RAN-REG transitions with latencies consistent with those expected from an ideal observer (*Barascud et al., 2016*; *Southwell et al., 2017*; *Southwell and Chait, 2018*).

In contrast, in Exp. 5, we demonstrated that longer term memory trace formation appears to require attentive processing in that there was no evidence for an immediate RT advantage when listeners moved from the exposure blocks, in which patterns were behaviourally irrelevant, to the active detection ('test') block. This suggests that the formation of lasting memories for sound patterns is not fully automatic, or does not immediately translate to behaviour. Whether this is driven by absence of attention per se or other factors is difficult to determine. For example, it is possible that the reduced memory effect when sounds are not behaviourally relevant is driven by decreased arousal or reward, known to substantially modulate learning (*Beste and Dinse, 2013*; *Braun et al., 2018*; *Polley, 2006*; *Yebra et al., 2019*), and which likely distinguish active detection (where feedback was provided after each trial) from passive listening.

Importantly, we showed that though implicit memory was not present at the onset of the test block, learning occurred more rapidly in the pre-exposed listeners, hinting at the presence of pre-exposure-related latent traces that may contribute to faster instantiation of representations in memory once the sequences become behaviourally relevant (*Frankland et al., 2019*).

### Conclusion

Uncovering how memory traces are encoded and preserved by the brain is crucial for understanding subsequent learning operations which drive pattern recognition and generalization. We showed that representations of sporadically reoccurring rapid sound patterns are retained in memory, thus facilitating detection when previously encountered patterns reoccur. In spite of the fact that the patterns were relatively featureless and undistinctive compared to real-world stimuli, this memory was robust, implicit, remarkably resistant to interruption, and persisted over long durations, revealing human listeners' astonishing sensitivity to reoccurring structure in the auditory environment. Important questions for future work include understanding the neurobiological foundations of these behavioural effects, the limits on the capacity of the memory store(s) involved and the factors which might affect subsequent forgetting.

## Materials and methods

### Stimuli

Stimuli were sequences of 50 ms tone-pips of different frequencies generated at a sampling rate of 44.1 kHz and gated on and off with 5 ms raised cosine ramps; the total sequence duration was 7 s (140 tones). Frequencies were drawn from a pool of twenty values logarithmically spaced between 222 and 2000 Hz; 12% steps. The order in which these frequencies were successively distributed defined different conditions that were otherwise identical in their spectral and timing profiles (see *Figure 1*). RAN sequences consisted of tone-pips arranged in random order, with the constraint that adjacent tones were not of the same frequency. Each frequency occurred equiprobably across the sequence duration. The RANREG condition contained a transition between a random (RAN), and a regularly repeating pattern (REG): sequences with initially randomly ordered tones changed into regularly repeating cycles of 20 frequencies (an overall cycle duration of 1000 ms; new on each trial). The change occurred between 3000 and 4000 ms after sequence onset such that each RANREG sequence contained between 3 to 4 REG cycles (only two in Exp. 2, see below). RAN and RANREG

conditions were generated anew for each trial and occurred equiprobably. Thus, each trial contained exactly the same frequency 'building blocks', with the same overall distribution, and only varied in the specific order of tone-pips. The inter-trial interval was jittered between 1400 and 1800 ms.

Unbeknown to participants, a few different REG patterns (different for each participant) reoccurred identically several times within the session (RANREGr condition). Reoccurrences happened three times per block (every ~3 min), and 9–15 times per session, depending on the number of blocks in the specific experiment. Note that the RAN part preceding each REGr was always novel. Reoccurrences were distributed within each block such that they occurred at the beginning (first third), middle and end of each block.

Two control conditions were also included within each block: sequences of tones of a fixed frequency (CONT), and sequences with a step change in frequency partway through the trial (STEP). The STEP condition served as a measure of individuals' reaction time to simple acoustic changes. The RT to STEP was subtracted from the RT to RANREG sequences to obtain a lower bound measure of computation time required to detect the transition.

Participants were instructed to respond, by pressing a keyboard button, as soon as possible after detecting a RANREG transition. Feedback about response accuracy and speed was delivered at the end of each trial. Since RT is a key performance measure in these experiments, it was important to motivate the participants to respond as quickly as possible. The feedback was given based on our previous work (*Barascud et al., 2016*), and consisted of a green circle if the response fell within 2200 ms from the regularity onset in the RANREG conditions, or within 300 ms from the change of tone in the STEP condition. For slower RTs, an orange circle (between 2200 and 2600 ms in the RANREG conditions, and between 300 and 600 ms in the STEP condition) or a red circle were displayed. It was explained to participants that they should strive to obtain as much 'green' or 'orange' feedback as possible. The experimental session was delivered in ~8 min blocks, separated by brief breaks. Stimuli were presented with PsychToolBox in MATLAB (9.2.0, R2017a) in an acoustically shielded room and at a comfortable listening level (self-adjusted by each listener).

## Participant numbers

We initially ran a pilot experiment (N = 20, 16 females, age 23.5 ± 2.95 years) which consisted of five consecutive blocks as in Exp. 1A. The effect size for the main effect of condition (RANREG vs. RANREGr) was $\eta_p^2$ = .48 and $\eta_p^2$ = .79 after the first 3 and 5 blocks respectively. Using $\eta_p^2$ = 0.48 for a prospective power calculation (beta = 0.8; alpha = 0.05) yielded a required sample size of N = 13. We decided to increase our sample size up to N = 20 to account for possible drop outs due to low accuracy. The research ethics committee of University College London approved the experiment [Project ID Number]: 1490/009, and written informed consent was obtained from each participant.

## Experiment 1a

The transition detection task was performed in three sessions: five blocks on day one ('day1'), one block after 24 hr ('24 hr') and another block after 7 weeks ('7 w'). Each block consisted of 60 stimuli (~8 min duration; 3 RANREGr x three reoccurrences per block, 18 RANREG, 27 RAN, 3 STEP, and 3 CONT). Feedback about the response accuracy and speed was delivered after each trial. Before starting, a short training block of 12 trials (with the same conditions as in the main experiment, but no RANREGr) was performed to acquaint participants with the task.

The familiarity task was performed at the end of each session (day1, 24 hr, 7 w). In these tests the three REGr patterns were randomly intermixed with 18 novel REG sequences. Participants were informed that a 'handful' of patterns reoccurred during the just completed session and asked to indicate, by means of a button press, if each presented pattern sounded 'familiar'.

Participants. Twenty paid individuals (ten females; average age, 24.4 ± 3.03 years) took part in the experiment. Because of poor accuracy (d' < 2 after the first block), one participant was excluded from the analysis. We were able to test only 14 participants after 7 weeks (eight females; average age, 24.7 ± 3.02 years). No participant reported hearing difficulties.

## Experiment 1b

Participants performed the transition detection task for six consecutive blocks consisting of the same set of stimuli described for Exp. 1A. In the 5th block, each REGr was time reversed.

Participants. Twenty paid individuals (13 females; average age = 24.25 ± 3.58 years) took part in the experiment. No participant reported hearing difficulties.

## Experiment 2

The stimulus set in the initial four blocks contained RANREG and RANREGr stimuli, as before, except they contained only two repeating cycles after the transition. To understand whether immediate repetition is necessary for memory to be formed two further conditions were used: PATinRAN stimuli contained two identical 20 tone patterns embedded amongst random tones (mean separation of 1.7 s; drawn randomly from a range. 5–2.9; the second appearance always occurred at the end of the sequence as shown in *Figure 3A*). Similar to REGr, three different PAT were designated as reoccurring across trials (different for each participant; three reoccurrences per block). The embedding RAN sequence and the spacing between the two PAT patterns were randomly set for each reoccurrence. Overall each block contained 82 stimuli (36 RAN, 9 RANREG, 9 RANREGr, 9 PATinRAN, 9 PATinRANr, 5 STEP, 5 CONT), with ISI between 2.4 and 2.8 s. Reoccurrences of RANREGr and PATinRANr occurred approximately every 3.6 min.

Participants were informed of the presence of PATinRAN and RANREG stimuli (but were naïve about RANREGr and PATinRANr) and were instructed to indicate, by button press, if they detected the presence of a repeating pattern within the just-heard sequence. Feedback was provided at the end of each trial as in the above experiments, except that in the PATinRAN conditions we delivered a green circle if the response fell within 1200 ms from the second cycle onset, a red circle if the response was slower that 1600 ms, and an orange one if it fell in between. It was explained to participants that they should be fast but prioritise accuracy, given the generally difficult level of the task.

In order to quantify any memory effects, in the 5th block ('test' block) each of PATinRANr sequences were replaced by sequences with the two cycles set adjacently. We will refer to this condition as RANREGr*. The test block contained 36 RAN, 18 RANREG, 9 RANREGr, 9 RANREGr*, 5 STEP, 5 CONT Stimuli were about 5.45 ms long (~109 tones).

Participants. Given the task complexity and expectation for a reduced SNR, we increased the number of participants, a-priori, by 50% relative to the previous experiment. Thirty paid individuals (twenty females; average age, 24.26 ± 3.8 years) took part in the experiment. No participants reported hearing difficulties.

## Experiment 3

This experiment consisted of two days of testing. On the first day participants performed a transition detection task as in Exp. 1A, but two different sets of reoccurring patterns (REGr1 and REGr2; three different patterns each) were presented. REGr1 was presented over the first three blocks, and REGr2 over the subsequent three blocks. On day 2 (after 24 hr), participants returned to the lab to perform two test blocks for the two sets of reoccurring regularities, REGr1 and REGr2 (order counterbalanced across participants).

Participants. We initially ran 20 participants (one excluded from analysis), but decided to run an additional 10 participants (+two excluded), to increase the SNR for the memory effects observed for the RANREGr1 and RANREGr2 conditions on day two. The results with N = 19 yielded qualitatively similar results (see *Figure 6—figure supplement 1B*). Thirty-two paid individuals (twenty females; average age, 24.5 ± 3.8 years) took part in the experiment. No participant reported hearing difficulties. Because of poor accuracy (d' < 2 after the first block), three participants were excluded from the analysis.

## Experiment 4

Participants performed the detection task through four consecutive blocks of 82 stimuli each. The stimulus set included the same conditions as described for Exp. 1A, but with six, instead of three, REGr sequences, each presented three times within a block (6 RANREGr x three reoccurrences per block, 18 RANREG, 36 RAN, 5 STEP, and 5 CONT). In block 4, REGr were phase shifted (see examples in *Figure 7A*). To ensure uniform sampling of possible phase shifts, for each REGr in block 4, each of the three intra-block presentations was subject to pattern phase shift of 2 to 7, 8 to 13, or 14 to 19 tones from the onset of the original pattern. The phase shift was determined independently for each REGr and each intra-block presentation. Stimulus duration was 6.5 s, and the transition time

was between 3 and 3.5 s from the sequence onset. Different REGr patterns reoccurred sparsely (every ~3.4 min) across trials and blocks.

Participants. Twenty paid individuals (fourteen females; average age, 23.5 ± 3.2 years) took part in the experiment. No participant reported hearing difficulties.

## Experiment 5

The experiment consisted of four blocks. The stimulus structure was as in Exp. 1A, except that for the first three blocks participants were instructed to respond to STEP changes only. They received no explanation about the regularity structure of the stimuli, and performed no practice. On the fourth block, they were instructed to detect RANREG transitions in addition to STEP transitions. Each block contained 72 stimuli (3 RANREGr x three reoccurrences per block, 18 RANREG, 27 RAN, 9 STEP, and 9 CONT; ISI between 900 and 1300); the number of STEP and CONT trials was increased relative to that in Experiment 1A due to the task change. As in Exp. 1A, participants performed the familiarity task at the end of the session.

Participants. Nineteen paid individuals (14 females; average age, 23.4 ± 3.1 years) took part in the experiment. No participant reported hearing difficulties. One participant was excluded from the analysis because of poor accuracy (d' < 1).

## Statistical analysis

In the transition detection task, two indexes of performance were computed: sensitivity (d') and reaction time (RT).

For each participant and each block, d' was quantified over trials (collapsed over RANREG and RANREGr) to give a general measure of sensitivity to the presence of regularities. Responses to RANREG and RANREGr, which occurred after the nominal transition were considered hits; Responses to RAN trials were considered false alarms. Participants who showed d' < 2 after the first block of the transition detection task were excluded from the analysis. Because Exp. five had only one 'active' block and no previous training, we adopted a more lenient exclusion criterion of d' < 1. Note that d' was not available in Exp. 2 because of the intermixed nature of the presentation of RANREG and PATinRAN stimuli. To quantify performance, we therefore focus on hit rates and false alarms. For the purpose of participant exclusion, we computed an overall d' (collapsing across conditions) and set the threshold at d' < 1.5.

Only RTs of correct trials (hits) were analysed. In all experiments, RT was defined as the time difference between the onset of the regular pattern ('nominal transition' in *Figure 1*) and the participant's button press. However, Exp. 2 contained conditions with non-contiguous pattern presentations. RT was therefore computed from the onset of the second cycle (as indicated in *Figure 3A*). Across all experiments, RTs which occurred before the transition to the regularity (see *Figure 1*; ~1.3% of the trials) were considered to indicate a false positive and excluded from the analysis. To control for individual latency of motor response to a simple acoustic change, RTs were then 'baselined' by subtracting the individual mean RT to the STEP transition. Moreover, for each participant and block, the RTs beyond 2 SD from the mean were discarded.

To quantify the formation of a memory trace over REGr presentations, RT were averaged to yield a mean RT per condition per subject per block. Therefore, RT to RANREGr were based on nine trials (3 REGr x three presentations per block). However, to evaluate the immediate presence of a memory trace following certain experimental manipulations (e.g. in Exp. 2 and 5) or when re-testing after 24 hr or 7 weeks (as in Exp. 1A) we also analysed RT for each intra-block presentation (the first, second and third intra-block instance of a REGr pattern; see *Figure 2—figure supplement 1*; *Figure 2—figure supplement 3*; *Figure 6—figure supplement 1A*; *Figure 7—figure supplement 1*; *Appendix 1—figure 1D*; *Appendix 1—figure 2D-J*; *Appendix 1—figure 3D*). To calculate the 'RT advantage' for each intra-block presentation, mean RTs of 1st, 2nd or 3rd intra-block presentation (averaged across the different REGr) were subtracted from the mean RTs of novel REG which occurred at the beginning (first third), middle or end of each block.

Performance was statistically tested with linear analyses of variance (ANOVA) implemented in the R environment (version 0.99.320) using the 'ezANOVA' function (*Michael Lawrence, 2016*). The analysis of d' modelled the repeated measures factor block (1: N blocks). The analysis on RTs modelled the repeated measures factors: condition (RANREG / RANREGr), block (1: N blocks), and their

interaction term. P-values were Greenhouse-Geisser adjusted when sphericity assumptions were violated. Post hoc t-tests were used to test for differences in performance between conditions across blocks or experiments. A Bonferroni correction was applied by multiplying p values by the number of comparisons (resulting values were capped at 1.0).

As a benchmark (see *Figure 9D*) across which to compare the effect of various manipulations on the RT advantage (i.e., *Figure 8D*, *Appendix 1—figure 2C-G*), we pooled data from several experiments to obtain a distribution of RT advantage values after each block: *Pooled data-block$_1$*, *Pooled data-block$_2$*, *Pooled data-block$_3$* were formed by pooling block 1, 2 or 3 data, respectively, from Experiments 1A, 1B, 3, 4, S1, S3, and pilot experiment identical to Exp. 1 (total N = 147). *Pooled data-block$_4$* was formed by pooling block four data from Experiments 1A, 1B, S1, S3 and the pilot (total N = 98), and *Pooled data-block$_5$* by pooling block 5 from Experiments 1A, S1 and the pilot (total N = 58). To obtain distributions of expected RT advantage values, data in each set were subjected to bootstrap resampling (1000 iterations) where, on each iteration, N random participants (N = number of participants in the experiment under examination) were drawn from the full pool, and their mean RT advantage (RANREG- RANREGr) was computed. This procedure yielded a distribution to which the actual data from the experiment under examination were compared. The p values provided (i.e., *Figure 8D*, *Appendix 1—figure 2C-G*) reflect the probability of the measured RT advantage (red dots in the relevant figures) relative to the benchmark distribution.

## Analysis of the familiarity task

The familiarity measurement required participants to categorize the presented patterns into 'familiar' (REGr) or 'new' (REG). Each REGr was presented once only, to avoid learning during the testing session and hence the 'familiar' category included only three items (six in Exp. S1, see Appendix1). These were presented among a larger set of foils (18 in Exp. 1A and Exp. 5, 36 in Exp. S1). Due to the small number of REGr, standard signal detection approaches are not useable. Instead, we computed the MCC score, which is a measure of the quality of a binary classification, applicable even when classes are of different sizes (*Boughorbel et al., 2017*; *Powers, 2007*). The coefficient ranges between 1 (perfect classification) to $-1$ (total misclassification) and is calculated using the following formula: $MCC = \frac{TP \times TN - FP \times FN}{\sqrt{[2](TP+FP)(TP+FN)(TN+FP)(TN+FN)}}$. Where TP = number of true positives; TN = number of true negatives; FP = number of false positives; FN = number of false negatives. The MCC scores obtained for each participant in Exp. 1A are shown in *Figure 2—figure supplement 2*.

## PPM-decay model

Prediction by Partial Matching (PPM) is a Markov modelling technique (*Cleary and Witten, 1984*) that models statistical structure within symbolic sequences by tabulating occurrences of *n*-grams within a training dataset. PPM is a variable-order Markov model, meaning that it generates predictions by combining *n*-gram models of different orders; here we use a model combination technique called 'interpolated smoothing' (*Bunton, 1996*; *Bunton, 1997*; see also *Pearce and Wiggins, 2004*; *Harrison et al., 2020*; for more details). This approach combines the advantages of both the structural specificity afforded by high-order n-gram predictions and the statistical reliability afforded by low-order n-gram predictions.

The PPM models used in prior cognitive research *Barascud et al., 2016*; *Cheung et al., 2019*; *Gold et al., 2019* have a 'perfect' memory, in that historic *n*-gram observations are preserved with the same fidelity as recent events, and are weighted the same in prediction generation. Noting that human memory exhibits clear capacity limitations and recency effects, *Harrison et al., 2020* modified PPM to incorporate a customizable decay kernel, whereby historic *n*-gram observations are down-weighted as a function of the time elapsed and the consequent *n*-grams observed since the initial observation. Modelling reaction-time data from a RANREG paradigm similar to *Barascud et al., 2016*, Harrison et al. concluded in favour of a capacity-limited high-fidelity echoic memory buffer followed by a lower-fidelity short-term memory phase with exponential decay. We likewise use an echoic-memory phase and a short-term memory phase in the present work, but add a slower-decaying long-term memory phase in order to capture the long-term learning observed in the present experiment.

The modelling aimed to reproduce behavioural performance qualitatively rather quantitatively. Many simplifications are made including that inter-sequence intervals, and breaks between

experimental blocks are modelled at a fixed rate of 1 s. We explored various parameter settings for the model, and retained the configuration that best reproduced the observed behavioural patterns in Experiments 1A, 2, and S2A (*Figure 5*, and *Appendix 1—figure 2K*), which represent the key manipulations of memory duration. The resulting parameters are listed in *Table 1*; the decay kernel is plotted in *Figure 4A*. Further implementation details are described in *Harrison et al., 2020*. The model outputs a conditional probability estimate for each tone in each sequence experienced throughout an experiment, which we convert to information content (the negative log probability in base 2). An implementation of this model is freely available in our open-source R package 'ppm' (https://github.com/pmcharrison/ppm; *Harrison, 2020*).

To identify changes in the information content profile corresponding to the RANDREG transition on a given trial, we use the nonparametric changepoint detection algorithm of *Ross et al., 2011*, which sequentially applies the Mann-Whitney test to identify changes in a time series' location while controlling for Type I error. Here, the target Type I error rate was set to 1 in 10000 tones. Note that, for simplicity, the change point detection algorithm is free of memory constraints. Human listeners likely use a rougher (less detailed) statistical representation for transition detection.

## Acknowledgements

We thank Barathy Ganeshakumara and Hera Ahmad for help with the behavioural data collection and Alain de Cheveigne for comments and discussion. This research was funded by a BBSRC grant (BB/P003745/1) to MC and supported by the NIHR UCLH BRC Deafness and Hearing Problems Theme.

## Additional information

### Funding

| Funder | Grant reference number | Author |
| --- | --- | --- |
| Biotechnology and Biological Sciences Research Council | BB/P003745/1 | Maria Chait |
| University College London Hospitals NHS Foundation Trust | Biomedical Research Centre Deafness and Hearing Problems Theme | Maria Chait |

The funders had no role in study design, data collection and interpretation, or the decision to submit the work for publication.

### Author contributions

Roberta Bianco, Conceptualization, Resources, Data curation, Software, Formal analysis, Supervision, Validation, Investigation, Visualization, Methodology, Writing - original draft, Writing - review and editing; Peter MC Harrison, Conceptualization, Resources, Software, Formal analysis, Validation, Investigation, Visualization, Methodology, Writing - review and editing; Mingyue Hu, Cora Bolger, Samantha Picken, Investigation, Writing - review and editing; Marcus T Pearce, Conceptualization, Resources, Supervision, Validation, Methodology, Writing - review and editing; Maria Chait, Conceptualization, Resources, Data curation, Formal analysis, Supervision, Funding acquisition, Validation, Methodology, Writing - original draft, Project administration, Writing - review and editing

### Author ORCIDs

Roberta Bianco https://orcid.org/0000-0001-9613-8933
Peter MC Harrison https://orcid.org/0000-0002-9851-9462
Maria Chait https://orcid.org/0000-0002-7808-3593

### Ethics

Human subjects: The research ethics committee of University College London approved the experiment, and written informed consent was obtained from each participant. Project ID Number: 1490/009.

Decision letter and Author response
Decision letter https://doi.org/10.7554/eLife.56073.sa1
Author response https://doi.org/10.7554/eLife.56073.sa2

## Additional files

### Supplementary files

- Supplementary file 1. Sound - RAN sequence.
- Supplementary file 2. Sound - RANREG sequence.
- Transparent reporting form

### Data availability

The datasets for this study can be found in the OSF repository: https://doi.org/10.17605/OSF.IO/DTZS3.

The following dataset was generated:

| Author(s) | Year | Dataset title | Dataset URL | Database and Identifier |
|---|---|---|---|---|
| Bianco R, Harrison P, Pearce M, Chait M | 2020 | Long-term implicit memory for sequential auditory patterns in humans | https://doi.org/10.17605/OSF.IO/DTZS3 | Open Science Framework, 10.17605/OSF.IO/DTZS3 |

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

## Appendix 1

## Experiment S2A, B: The memory trace is weakened, but not abolished by interrupting blocks

Although reoccurrence of regularities was quite sparse in Exp. 1A (every ~2.7 min), they were presented regularly over five blocks. Here, we asked whether memory formation can be interrupted by introducing a delay of 10 min ('interrupting blocks' in which REGr were not presented) between 'standard blocks'.

Methods: These experiments involved the same transition detection task as in Exp. 1A, but 'interrupting blocks', in which RANREGr condition was not presented, were introduced between 'standard blocks'. The 'interrupting blocks' were block 2 and 4 in experiment S2A, block 3 and 5 in experiment S2B. Across five blocks, in experiment S2A participants were presented with 27 RANREGr, 108 RANREG, 135 RAN, 15 STEP, and 15 CONT. Across six blocks, in experiment S2B participants were presented with 36 RANREGr, 126 RANREG, 162 RAN, 18 STEP, and 18 CONT.

Participants of experiment S2A. Nineteen paid individuals (13 females; average age, 23.8 ± 4.7 years) took part in the study. No participant reported hearing difficulties. Because of poor accuracy (d' < 2 after the first block), one participant was excluded from the analysis.

Participants of experiment S2B. Twenty paid individuals (10 females; average age, 23.8 ± 4.00 years) took part in the study. No participant reported hearing difficulties. Because of poor accuracy (d' < 2 after the first block), one participant was excluded from the analysis.

Results: In Exp. S2A, an interrupting block was inserted after each standard block (*Appendix 1—figure 2B*). The RT data demonstrated an RT advantage to reoccurring vs. novel regularities (~130 ms – 2.6 tones by the end of the third standard block), which did not improve substantially between the second and third standard blocks [main effect of condition: $F(1, 17) = 35.03$, $p < 0.001$, $\eta_p^2 = .67$; main effect of block: $F(2, 34) = 10.67$, $p < 0.001$, $\eta_p^2 = .39$; no interaction: $F(2, 34) = 3.03$, $p = 0.061$, $\eta_p^2 = .15$]. The RT advantage here was smaller than that typically observed after three consecutive blocks (~180 ms – 3.7 tones in *Pooled data-block₃*; *Appendix 1—figure 2C*; difference significant at $p = 0.027$ based on bootstrap resampling; see Materials and methods in the main document).

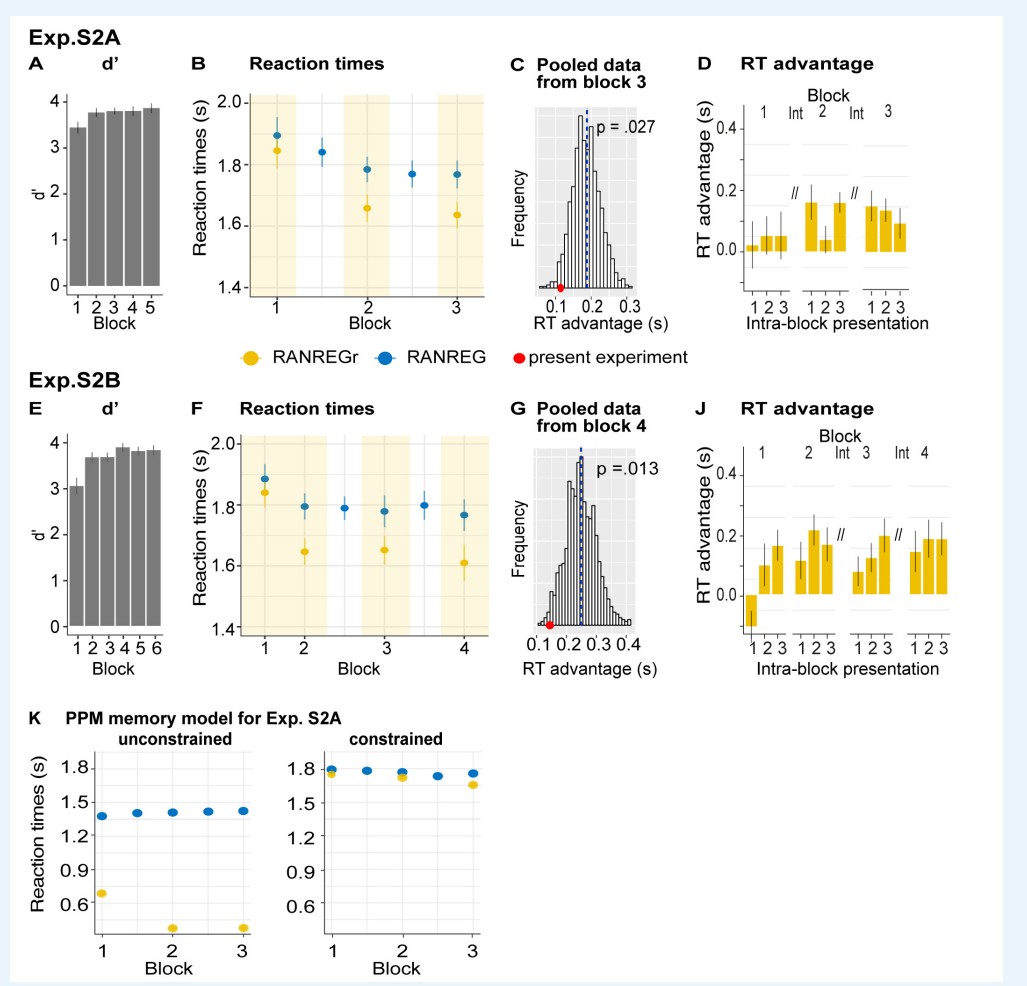

**Appendix 1—figure 2.** Experiment S2A (N = 18) and S2B (N = 19): the memory trace is weakened, but not abolished, by interrupting blocks. (**A–D**) Exp. S2A: (**A**) Sensitivity to emergence of regularity (d') across blocks in experiment S2A. Error bars indicate 1 s.e.m. (**B**) RTs to transition in RANREG and RANREGr across blocks. Error bars indicate 1 s.e.m. Yellow shading indicates blocks where REGr were present. (**C**) Bootstrap resampling-based distributions of RT advantage after three uninterrupted blocks (*Pooled data-block₃*; see Materials and methods). The red dot indicates the RT advantage measured after block three in the present experiment. (**D**) RT advantage for each intra-block presentation. The RT advantage was preserved over 'interrupting' blocks. Plotted values correspond to the RT advantage of REGr for each intra-block presentation. RTs of 1st, 2nd or 3rd intra-block presentations were averaged across the different REGr, and RTs to novel REG were averaged across trials which occurred at the beginning (first third), middle or end of each block. Note that the RT for REGr is computed based on three trials and the effects are therefore rather noisy. Error bars indicate 1 s.e.m. (**E–J**) Exp. S2B: (**F**) Sensitivity to emergence of regularity (d') across blocks for experiment S2B Error bars indicate 1 s.e.m. (**F**) RTs to the transition in RANREG and RANREGr across blocks. Error bars indicate 1 s.e.m. Yellow shading indicates blocks where REGr were present. (**G**) Bootstrap resampling-based distributions of RT advantage after 4th blocks (*Pooled data-block₄*; see Materials and methods). The red dot indicates the RT advantage measured after block four in the present experiment. (**J**) The RT advantage was preserved over 'interrupting' blocks. (**K**) Unconstrained vs. Constrained memory model results for Exp. S2A. Error bars indicate 1 s.e.m.

In Experiment S2B, we introduced the first interrupting block after block two in order to allow for the memory trace to emerge (see *Appendix 1—figure 2F*). The RT advantage in the

2nd block was similar to that observed in the control (*Pooled data-block$_2$*: p=0.48), but no considerable improvement was observed across blocks thereafter [main effect of condition: F (1, 18) = 74.93, p < 0.001, $\eta_p^2$ = .81; main effect of block: F(3, 54) = 11.19, p < 0.001, $\eta_p^2$ = .38; no interaction: F(2, 54) = 2.56, p = 0.064 $\eta_p^2$ = .12]. The RT advantage in the blocks thereafter was indeed smaller than under 'uninterrupted' control conditions (block three vs. *Pooled data-block$_3$*: p = 0.071; block four vs. *Pooled data-block$_4$*: p = 0.013, see **Appendix 1—figure 2G**).

These results suggest that the memory trace for REGr can withstand quite substantial interruptions: suspending the regular reoccurrences of REGr (by introducing 'interrupting blocks') resulted in a largely maintained memory, though there was evidence for a somewhat stagnated RT advantage.

Modelling Exp. S2A. The performance of the unconstrained PPM model (**Appendix 1—figure 2K**), was not affected by the interruptions (also compare this figure with **Figure 5**-A in the main text). In contrast, in the memory-decay PPM model inserting 'interrupting' blocks has the effect of reducing the memory traces of previously heard regularities. The constrained model shows somewhat worse performance relative to the constrained model in Exp. 1A, consistent with human effects.

## Experiment S3: Implicit memory is robust to pattern transposition

We tested whether the implicit memory for reoccurring sequences generalises to versions in which relative relationships within the stimulus (pitch intervals) are preserved, while absolute information (the frequency values themselves) are manipulated.

Methods: The stimulus set included the same conditions as described for Exp. 1A, but with the following differences: RAN sequences were generated from a pool of twenty-six frequencies (logarithmically-spaced values between 222 and 4004 Hz; 12% steps). REG patterns consisted of 20 frequencies randomly selected from the pool. To allow for the transposition, REGr patterns were drawn from a subset of 24 frequencies (i.e., not including the highest and lowest frequency in the pool). In the 5$^{th}$ block, each REGr was randomly transposed up or down by one tone (12%; shifted one place higher or lower in the frequency pool than the original, see 3-A).

Participants. Twenty paid individuals (twelve females; average age, 24.75 ± 6.8 years) took part in the study. No participant reported hearing difficulties.

Results: Overall, the same pattern of performance as in Exp. 1A was seen. **Appendix 1—figure 3C** demonstrates progressively stronger implicit memory for REGr, as revealed by a growing RT advantage over novel REG across blocks [main effect of condition: F(1, 19) = 47.31, p < 0.001, $\eta_p^2$ = .71; main effect of block: F(4, 76) = 7.95, p < 0.001, $\eta_p^2$ = .29; interaction condition per block: F(4, 76) = 5.35, p = 0.003, $\eta_p^2$ = .22]. Specifically, whilst in the first block performance did not differ between RANREG and RANREGr conditions [t (19) = 1.635, p = 0.59], a significantly faster response (186 ms; 3.7 tones) for RANREGr was observed in the second block [t(19) = 4.302 p = . 001], and grew across the remaining blocks (all ps < 0.004).

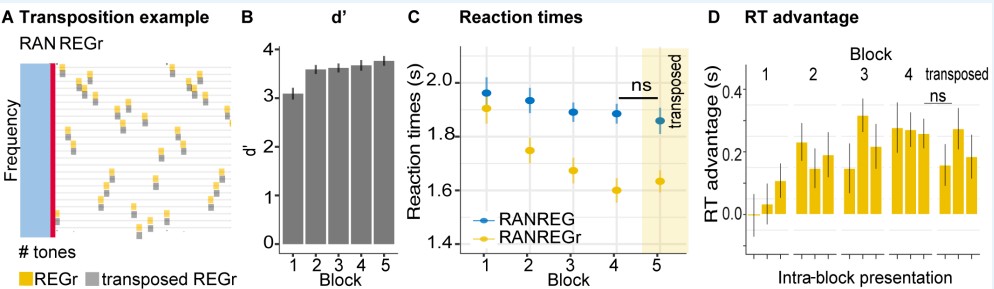

**Appendix 1—figure 3.** Experiment S3 (N = 20): implicit memory is robust to pattern transposition. (**A**) Schematic example of the transposition. Yellow squares indicate tones in a REGr sequence; grey squares indicate the transposed version (in this example, the tones were

shifted downwards by one step in the frequency pool; 12%). The red line indicates the transition from RAN (light blue area) to REGr. (**B**) d' across all blocks. Error bars indicate 1 s. e.m. (**C**) RT to the transition in RANREG and RANREGr across blocks. In block 5 (yellow shading) the originally learned REGr were replaced by transposed versions. Error bars indicate 1 s.e.m. (**D**) RT advantage for each intra-block presentation. The RT advantage was preserved following frequency transposition of the REGr pattern. Plotted values correspond to the RT advantage of REGr for each intra-block presentation. RTs of 1st, 2nd or 3rd intra-block presentations were averaged across the different REGr, and RTs to novel REG were averaged across trials which occurred at the beginning (first third), middle or end of each block. Note that the RT for REGr is computed based on three trials and the effects are therefore rather noisy. Error bars indicate 1 s.e.m.

Importantly, this RT advantage (205 ms – 4.1 tones) in block 5 (transposed REGr) did not differ from the RT advantage on block 4 (272 ms; 5.4 tones) [t(19) = 1.541, p = 0.14]. To confirm the immediacy of the transfer we compared the RT advantage in the first intra-block presentation in block 5, where the transposition was introduced, with the third (last) intra-block presentation in block 4 (*Appendix 1—figure 3D*). No difference was observed [t (19) = 1.26, p = 0.223], suggesting that the generalization to the transposed pattern was instantaneous.

The observation of a transfer of RT advantage to the transposed sequences may suggest that the formed representation is not precisely echoic: instead of the specific frequency pattern, the auditory system might be maintaining a representation of the contour, or inter-tone interval within the REGr pattern. Another possibility is that the tolerance reflects a noisy frequency representation, though we note that the frequency steps here (12%) are large enough to be discriminable by most listeners.

