## [Decision Letter]

**Acceptance summary:**

This paper shows, in our view very convincingly, that human listeners rapidly form robust and surpirisingly long-lasting memories of rarely encountered, featureless sound sequences presented among many similar stimuli. Using a previously established listening paradigm of such sound sequences and employing a compelling set of control conditions and analyses, including an ideal-observer mode, these findings are surprising and exciting. The experiments are well controlled and the analysis considers alternate hypotheses. Subsequent experiment answer additional questions to fully characterize the effect and demonstrate that it is acoustic-frequency invariant, long-lasting, and resistant to interference.

The paper connects to a long tradition in psychoacoustics using such tone patterns as stimuli. However, the huge majority of those studies focused on sensory memory or short-term learning. We think that the current research opens a new avenue by shifting the focus towards long-term memory for tonal patterns. While the main finding is sufficiently robust and compelling to be published, the results are also likely to provoke rich follow-up research. With its elegantly designed experiments using behavioral measures (accuracy, reaction times) as dependent variables and including computational modeling, the paper establishes a foundation for future neuroscientific studies.

**Decision letter after peer review:**

Thank you for submitting your article "Long-term implicit memory for sequential auditory patterns in humans" for consideration by *eLife*. Your article has been reviewed by Barbara Shinn-Cunningham as the Senior Editor, a Reviewing Editor, and three reviewers. The following individuals involved in review of your submission have agreed to reveal their identity: Erich Schroger (Reviewer #2); Kasia Bieszczad (Reviewer #3).

The reviewers have discussed the reviews with one another and the Reviewing Editor has drafted this decision to help you prepare a revised submission.

Summary:

The reviews yielded a surprisingly large degree of overlap: This manuscript describes an unusually long-lasting implicit memory of sound sequence based on repetitive exposure alone. Within it, the authors describe time constants that constrain the formation of this memory that are modeled to replicate the human data. Overall, these findings are surprising and exciting.

The usage of such tonal patterns has a long tradition in psychoacoustics (since the 1970s, e.g. Watson.et al., 1975) and cognitive psychophysiology (since the 1990s, e.g. Schröger, Näätänenand Paavilainen, 1992) to study auditory processing in humans (and animals) still is a very timely topic resulting in hundreds of papers. However, the huge majority of those studies focused on sensory memory or short-term learning (with some exceptions also studying long-term learning of tonal patterns such as Atienza and Cantero, 2001; Näätänen et al., 1993). The current research by Biancoand et al., opens a new avenue by shifting the focus towards long-term memory for tonal patterns. Bianco and colleagues present a series of elegantly designed experiments using behavioral measures (accuracy, reaction times) as dependent variables.

The experiments are well-controlled and with excellent consideration of alternate hypotheses. We found answers to arising questions in subsequent experiments throughout the manuscript that fully characterized the effect to be acoustic-frequency invariant, long-lasting and resistant to interference.

The work is well done, well written, and interesting. As a grain of salt, some of us were missing the bigger conclusion that can be drawn from the work beyond the individual experiments. We identified three revision-worthy issues: conceptual (all three reviewers); modeling (reviewers 1 and 2); and methods in general (see below for details).

Essential revisions:

1) All reviewers and editors agreed that the findings must be couched better in the existing learning/memory literature w.r.t. MMN and exposure-only learning models in adult brains (e.g., songbirds). As for the modeling, we agreed agree that the modeling could be better integrated with the behavioral findings in a way that is additive rather than somewhat circular. In other words, what does the modeling experiment actually contribute to the discovery? Does it suggest mechanisms, constraints?

2) Abstract and conclusions: The abstract could tell the reader more clearly what we learn from this paper. The abstract describes what was done and what the results are, but why this question is important and what to conclude from the work remains not that clear (the last sentence feels opaque and not fully related to the data; or at least this could be clarified more). A conclusion section is missing entirely. We urge the authors to add such conclusion, in order to wrap up with clear statements (but clearly related to the data) what the reader may take from this study. This will help increase the lasting impact of this work.

3) A relatively big concern relates to the modeling. The authors may want to emphasize more strongly what new the reader learns from the modeling. We do know that memory decays (so the unconstrained model is not very interesting). The exact time course of decay is not very interesting as this will dependent on the stimulus and paradigm, and will likely be different for different stimuli. The model does not explain the longer-term effects well (e.g., 7w). The modeling section in the results reflects largely a re-description of the experimental data. The model was not fit to the individual participant data, nor was it compared to an alternative model. It is not necessarily a problem that a descriptive model is presented to explain some hypothetical mechanism, but at the same time, the model (n-grams) seems to work particularly with discrete countable events (such as the tones used here). We wondered how much it generalizes. The impression occured that the model was "made to look like" the data, which ends up being circular. One might even consider removing it, since the experimental data speak for themselves, but we assume the authors are more married to the model. We thus recommend making clear what we learn anew from the model that is not known from the literature and that is generalizable beyond the experimental design utilized here.

4) Analysis and plotting related to Figure 8C/D: We recommend the authors include the data from the first block (pooled data) in Figure 8C (i.e., RANREG and RANREGr for each intra-block). This would provide an additional useful visualization of the data (this could also be accomplished using their bootstrapping approach: for each iteration, the mean and variance could be stored; mean and variance are then averaged across iterations and the SEM is calculated based on the variance [with N=36]). Moreover, the authors may also want to provide results from more a traditional analysis: an ANOVA with RT advantage for 1-3 intra-blocks (within-subjects) and group (between-subjects). It is not a problem that there are differences in the number of the available participants per group.

5) We strongly recommend removing the separation into 25% (low performers) and 75% (high performers). This seems unmotivated conceptually, the data distribution is normal (or at least uni-modal) and thus does not justify dividing the distribution arbitrarily, leads to circular wording (e.g., that low performers do worse [put simply]), and may lead to speculations (e.g., "what cognitive abilities distinguish the low- from the high-performers", subsection “Across-experiment analysis reveals that most patterns are remembered and most participants exhibit implicit memory”; it could also be differences in motivation, sensory abilities, or any number of variables; moreover, statistically speaking, there will always be a 25% bottom group for a uni-modal distribution).

6) The authors may want to acknowledge explicitly that their passive condition is actually active with respect to the auditory stimulation. The data are not fully conclusive, in my view, whether passive (as in not actively listening to the sounds) leads to similar behavioral patterns. I think this distinction needs to be addressed more clearly in the results and discussion.

7) Authors should consider contrasting effects to the relevant birdsong literature that also describes statistical learning and its underlying neurobiological mechanisms

8) Have the authors considered memory consolidation effects to explain time-dependent processes that oppose memory decay? These are known to improve performance after an incubation period (time passing) without continued exposure or training. This may be relevant to explain the maintenance of memory after >7 weeks.

---

## [Author Response]

Essential revisions:1) All reviewers and editors agreed that the findings must be couched better in the existing learning/memory literature w.r.t. MMN and exposure-only learning models in adult brains (e.g., songbirds). As for the modeling, we agreed agree that the modeling could be better integrated with the behavioral findings in a way that is additive rather than somewhat circular. In other words, what does the modeling experiment actually contribute to the discovery? Does it suggest mechanisms, constraints?

Thank you for these suggestions. We now expanded the Discussion section to refer to previous MMN work (Subsection “Memory for auditory sequences”) and findings from the song bird literature (subsection “Time scales of memory for sequences”). We also emphasize the usefulness of the model (for details see reply to question 3 below).

Subsection “Memory for auditory sequences” (MMN and exposure only learning models):

‘Signals based on tone-pip patterns have long been used to understand how auditory memory affects human listeners’ perception of sound sequences (e.g. Watson, Wroton, Kelly, and Benbassat, 1975; Atienza and Cantero, 2001; Näätänen, Schröger, Karakas, Tervaniemi, and Paavilainen, 1993; Schröger, Näätänen, and Paavilainen, 1992; Tervaniemi, Rytkönen, Schröger, Ilmoniemi, and Näätänen, 2001; Moldwin, Schwartz, and Sussman, 2017). However, these paradigms are predominantly based on extensive exposure (in the order of hundreds of consecutive repetitions) to a single pattern.’

Subsection “Time scales of memory for sequences” (Songbirds literature):

‘In animal models, repetitive exposure to sound tokens (though, notably at a much higher rate than that used here) has been shown to evoke a process of long-lasting adaptation manifested as sparser firing and increased response specificity. These effects, persisting for hours to days after the initial stimulation, have been observed in primary and secondary auditory areas in song birds (Caudal Medial Nidopallium; Cazala, Giret, Edeline, and Del Negro, 2019; Honda and Okanoya, 1999; Lu and Vicario, 2014; Menyhart, Kolodny, Goldstein, DeVoogd, and Edelman, 2015; Takahasi et al., 2010; Chew, Vicario, and Nottebohm, 1996; Soyman and Vicario, 2019) and in secondary auditory cortex in ferrets (Lu et al., 2018). The hypothesis that similar processes might back the behavioural effects we report is appealing.’

2) Abstract and conclusions: The abstract could tell the reader more clearly what we learn from this paper. The abstract describes what was done and what the results are, but why this question is important and what to conclude from the work remains not that clear (the last sentence feels opaque and not fully related to the data; or at least this could be clarified more). A conclusion section is missing entirely. We urge the authors to add such conclusion, in order to wrap up with clear statements (but clearly related to the data) what the reader may take from this study. This will help increase the lasting impact of this work.

We changed the Abstract as follows:“Memory, on multiple timescales, is critical to our ability to discover the structure of our surroundings, and efficiently interact with the environment. We combined behavioural manipulation and modelling to investigate the dynamics of memory formation for rarely reoccurring acoustic patterns. In a series of experiments, participants detected the emergence of regularly repeating patterns within rapid tone-pip sequences. Unbeknownst to them, a few patterns reoccurred every ~3 minutes. All sequences consisted of the same 20 frequencies and were distinguishable only by the order of tone-pips. Despite this, reoccurring patterns were associated with a rapidly growing detection-time advantage over novel patterns. This effect was implicit, robust to interference, and persisted up to 7 weeks. The results implicate an interplay between short (a few seconds) and long-term (over many minutes) integration in memory formation and demonstrate the remarkable sensitivity of the human auditory system to sporadically reoccurring structure within the acoustic environment.”

We added a conclusion paragraph that reads:

“‘Uncovering how memory traces are encoded and preserved by the brain is crucial for understanding subsequent learning operations which drive pattern recognition and generalization. […] Important questions for future work include understanding the neurobiological foundations of these behavioural effects, the limits on the capacity of the memory store(s) involved and the factors which might affect subsequent forgetting.”

3) A relatively big concern relates to the modeling. The authors may want to emphasize more strongly what new the reader learns from the modeling. We do know that memory decays (so the unconstrained model is not very interesting). The exact time course of decay is not very interesting as this will dependent on the stimulus and paradigm, and will likely be different for different stimuli. The model does not explain the longer-term effects well (e.g., 7w). The modeling section in the results reflects largely a re-description of the experimental data. The model was not fit to the individual participant data, nor was it compared to an alternative model. It is not necessarily a problem that a descriptive model is presented to explain some hypothetical mechanism, but at the same time, the model (n-grams) seems to work particularly with discrete countable events (such as the tones used here). We wondered how much it generalizes. The impression occured that the model was "made to look like" the data, which ends up being circular. One might even consider removing it, since the experimental data speak for themselves, but we assume the authors are more married to the model. We thus recommend making clear what we learn anew from the model that is not known from the literature and that is generalizable beyond the experimental design utilized here.

We rewrote the relevant sections so as to emphasize the usefulness of the model. In particular, we highlight the following:

- Although the existence of memory decay in humans is in general well established, ways of incorporating memory decay into probabilistic computational models of sequences processing is very much an active topic of research.

- Though it has many constraints and simplifications, the model is useful in concretizing the effect of interplay between short- and longer- time scales in the formation of enduring memories for sequences.

- In particular, the biologically plausible set of parameters fully account for the dynamics of memory formation over reoccurrences.

- The insight into possible single-trial level dynamics afforded by the modelling can be useful for constraining the search for the neural underpinnings of the observed effects.

- We also acknowledge (subsection “Modelling”) that although our model does not explain our long-term effects, to our knowledge there is no other statistical learning model that accounts both for learning dynamics and long-term fixed effects.

- The unconstrained model has been used successfully in previous research (Barascud et al., 2016) to help understand the responses of listeners. In the present paper, the comparison with the non-constrained model is useful for providing a benchmark – i.e. demonstrating the effect the change in parameter has on model output.

Subsection “Modelling”: ‘We constructed a ‘memory constrained’ computational model, based on ‘prediction by partial matching’ (PPM; see Materials and methods section) to provide a formal simulation of the psychological mechanisms underlying the process of memory trace formation, as observed in Experiments 1A (Figure 2), 2 (Figure 5) and S2A (Appendix1—figure 2.Figure D). These experiments reflect critical manipulations of the effect of long- and short- term memory decay. Although the existence of memory decay in humans is in general well established, ways of incorporating memory decay into probabilistic computational models of sequences processing is very much an active topic of research. Our PPM model implemented a single set of values (Table 1) that fully accounted for the dynamics of memory formation observed across experiments. As a benchmark, we also report the results for an equivalent unconstrained model (i.e., with perfect memory), as employed in previous research using the same paradigm (Barascud et al., 2016). The following cognitive hypotheses were instantiated: (1) Listeners learn sequence transition probabilities throughout the experiment. This approach is similar to other models of statistical learning (Bröker, Bestmann, Dayan, and Marshall, 2018; Harrison, Bestmann, Rosa, Penny, and Green, 2011; Meyniel, Maheu, and Dehaene, 2016a; Takahasi, Yamada, and Okanoya, 2010) except the present model extends beyond first-order transition probabilities. Learning of sequence statistics is accomplished though partitioning the unfolding stimulus into sub-sequences of increasing order (n-grams) that are thereon stored in memory, such that the more a listener is exposed to a given n-gram, the stronger its salience (‘weight’). Here, we allow n to range between 1 and 5, corresponding to Markovian transition probabilities of orders 0 to 4. ….’

Subsection “Modelling”: “Overall, the modelling successfully replicated the slow dynamics of memory formation exhibited by human listeners demonstrating that a memory constrained transition-probability learning is a plausible computational underpinning of sequential pattern acquisition.”

We also discuss model limitations and possible generalizability to natural sounds. The subsection “What is being remembered?”Discussion section now reads:

“Similar to other models of statistical learning (Bröker et al., 2018; Harrison et al., 2011; Meyniel, Maheu, and Dehaene, 2016b), our memory-constrained PPM model explicitly assumes that listeners represent the unfolding sequences in the form of n-gram sub-sequences of variable length, from which transition probabilities are computed. Previous computational, behavioural and neuroimaging studies (Bianco, Ptasczynski, and Omigie, 2020; Conklin and Witten, 1995; Di Liberto et al., 2020; Egermann, Pearce, Wiggins, and McAdams, 2013; Pearce, Ruiz, Kapasi, Wiggins, and Bhattacharya, 2010; Pearce and Wiggins, 2004, 2006) demonstrated that PPM successfully generalizes to prediction of musical sequences and effectively accounts for psychophysiological responses to melodies. In particular, PPM provided a good match to brain response latencies evoked by transitions between RAN and REG patterns (Barascud et al., 2016; Southwell and Chait, 2018), suggesting that listeners may rely on similar memory representations as those proposed by the model. Here, the memory constrained version of PPM was able to successfully simulate human performance – concretizing how the interplay between short- and long- term decay might give rise to the progressive emergence of a memory trace across presentations. Whether listeners do indeed represent auditory patterns in this way is a matter of ongoing debate (e.g. Thiessen, 2017). Additional support for an n-gram-like representation is provided in Exp. 4, which demonstrated that the REGr RT advantage is robust to pattern phase shifts. This finding indicates that REG patterns are not encoded in memory as rigid chunks of sequential items (Perruchet and Pacton, 2006; Thiessen, 2017), but are instead represented as a transition rule which allows for flexible retrieval. Whilst further empirical evidence is essential to determine the nature of the memory representation, the insight into single-trial level dynamics derived from the present modelling (Figure 4) may be useful for constraining the search for the physiological underpinnings of these phenomena. Furthermore, the model can readily be applied to statistical learning in other modalities (reviewed by Frost et al., 2019) and even in other species, including songbirds such as finches, known to be capable of statistical learning (Menyhart et al., 2015; Takahasi et al., 2010).

A related question pertains to the generalizability of the present model to natural sounds beyond quantized sequences, such as those used here. In order to relate listeners’ performance to a measure of statistical information within unfolding signals, simplifying assumptions are necessary. This includes the presence of a prior stage of category formation which converts a continuous sound into discrete units that form the model’s ‘alphabet’. Accumulating evidence is indeed revealing that unsupervised segmentation of unfolding sounds into basic elements, perhaps using envelope-based cues, may be an inherent feature of listening (Ding, Melloni, Tian, and Poeppel, 2017; Doelling, Arnal, Ghitza, and Poeppel, 2014; Hickok and Poeppel, 2007; Poeppel, 2003).”

We have also tightened the subsection “Modelling” to avoid redundancy with the behavioural data section. Unfortunately, we cannot model individual participants because the implicit reaction-time paradigm is rather noisy and can only deliver a good signal when we average over participants.

4) Analysis and plotting related to Figure 8C/D: We recommend the authors include the data from the first block (pooled data) in Figure 8C (i.e., RANREG and RANREGr for each intra-block). This would provide an additional useful visualization of the data (this could also be accomplished using their bootstrapping approach: for each iteration, the mean and variance could be stored; mean and variance are then averaged across iterations and the SEM is calculated based on the variance [with N=36]). Moreover, the authors may also want to provide results from more a traditional analysis: an ANOVA with RT advantage for 1-3 intra-blocks (within-subjects) and group (between-subjects). It is not a problem that there are differences in the number of the available participants per group.

Thank you for the suggestions. We have:

1) Included a direct comparison with the control group (N=147) using traditional analysis (t-test).

2) Within the pre-exposed group we quantified RT advantage for each intra-block presentation and compared to 0 as a measure of the presence of a memory effect.

We also ran a between-group ANOVA with intra-block repetition as within factor on the RT advantage, as suggested by the reviewer. We found a main effect of group [F(1,160) = 8.60, p = .004, ges = .021], a main effect of intra-block presentation [F(2,308) = 7.45, p <.001, ges = .027, but not in interaction with group [F(2,308) = 1.11, p = .329, ges = .004] (ges = generalised eta square). This indicates that RT advantage grows across intra-blocks presentation in both groups, but the overall RT advantage achieved by the pre-exposed group is greater than that of the ‘control’ group. These results are entirely consistent with the bootstrap analysis reported originally and are therefore not included in the text to reduce redundancy. We believe the non-parametric bootstrap analysis is the more robust approach for this sort of analysis.

The relevant part in subsection “Experiment 5: Implicit memory can form when sounds are behaviourally irrelevant, but does not immediately transfer to behaviour” now reads:

“We analysed the performance in the test block of the pre-exposed group in comparison to the performance of a non pre-exposed ‘control’ group, formed by pooling block 1 data from several other experiments (Pooled data-block1, N = 147, see Materials and methods section). ….

In the test block (Figure 8B), the mean RT to RANREGr was significantly faster than that to novel RANREG [t(17) = 3.1, p = 0.006], consistent with the presence of an RT advantage. The RT advantage in the pre-exposed group (~157 ms, 3.14 tones) was substantially greater than in the control group (~30 ms, 0.6 tones) [independent sample t(163) = 3.023, p = .003], indicating a beneficial effect of pre-exposure.

As a critical test for the presence of a memory trace after pre-exposure, we examined RT in each intra-block presentation of REGr. If memories for reoccurring patterns are formed during pre-exposure, an RT advantage should be exhibited immediately – at the first presentation of REGr in the test block. One sample t-tests demonstrated that an RT advantage was absent at the first and second intra-block presentations [t(16) = .377, p = .711; t(17) = 1.691, p = .109], but emerged at third presentation of REGr [t(17) = 3.954, p = .001]. We also compared the RT advantage, across intra-block presentations between the pre-exposed and control groups. A bootstrap approach (see Materials and methods section) was used to generate a distribution of performance over subsets of 20 participants drawn from the control group and to compare with the actually observed performance in the pre-exposed group (Figure 8D).”

5) We strongly recommend removing the separation into 25% (low performers) and 75% (high performers). This seems unmotivated conceptually, the data distribution is normal (or at least uni-modal) and thus does not justify dividing the distribution arbitrarily, leads to circular wording (e.g., that low performers do worse [put simply]), and may lead to speculations (e.g., "what cognitive abilities distinguish the low- from the high-performers", subsection “Across-experiment analysis reveals that most patterns are remembered and most participants exhibit implicit memory”; it could also be differences in motivation, sensory abilities, or any number of variables; moreover, statistically speaking, there will always be a 25% bottom group for a uni-modal distribution).

Done (Figure 9). In line with the suggestion, we also removed that separation from Figure 7.

6) The authors may want to acknowledge explicitly that their passive condition is actually active with respect to the auditory stimulation. The data are not fully conclusive, in my view, whether passive (as in not actively listening to the sounds) leads to similar behavioral patterns. I think this distinction needs to be addressed more clearly in the results and discussion.

This is a very good point, that we already explain in the text, but the title is misleading. We changed the wording to ‘behaviourally irrelevant’ instead of ‘passive’ (subsection “Experiment 5: Implicit memory can form when sounds are behaviourally irrelevant, but does not immediately transfer to behaviour”Figure 8).

7) Authors should consider contrasting effects to the relevant birdsong literature that also describes statistical learning and its underlying neurobiological mechanisms

Done (Figure 9). In line with the suggestion, we also removed that separation from Figure 7.

Thank you for this suggestion. We have now added an explicit discussion of relevant findings from the birdsong literature (subsection “Time scales of memory for sequences”). which now reads:

“In animal models, repetitive exposure to sound tokens (though, notably at a much higher rate than that used here) has been shown to evoke a process of long-lasting adaptation manifested as sparser firing and increased response specificity. These effects, persisting for hours to days after the initial stimulation, have been observed in primary and secondary auditory areas in song birds (Caudal Medial Nidopallium; Cazala, Giret, Edeline, and Del Negro, 2019; Honda and Okanoya, 1999; Lu and Vicario, 2014; Menyhart, Kolodny, Goldstein, DeVoogd, and Edelman, 2015; Takahasi et al., 2010; Chew, Vicario, and Nottebohm, 1996; Soyman and Vicario, 2019) and in secondary auditory cortex in ferrets (Lu et al., 2018). The hypothesis that similar processes might back the behavioural effects we report is appealing.”

8) Have the authors considered memory consolidation effects to explain time-dependent processes that oppose memory decay? These are known to improve performance after an incubation period (time passing) without continued exposure or training. This may be relevant to explain the maintenance of memory after >7 weeks.

Thank you for this interesting suggestion. We agree that the account provided by the reviewers is a possible one, but since we are dealing with ‘coarse’ behavioural data it is difficult to strongly claim memory consolidation processes over very slow decay. We refer to this point in the sections below:

“It is important to note that the steady long-term decay, which is a key feature of the memory constrained model predicts that the performance facilitation should disappear after 24 hours, and certainly after 7 weeks. After such time periods, the memory traces for the reoccurring patterns should decay to zero, and the corresponding facilitation effect should disappear. Remarkably, the participants exhibited unaltered performance facilitation. This suggests that the memory traces of these reoccurring patterns are somehow ‘fixed’ at a certain point during testing. One way of simulating this effect would be to change the asymptote of the exponential memory decay, such that the memory trace asymptotically approaches a small but non-zero value as time tends to infinity. However, we found that incorporating such an asymptote caused the performance facilitation for RANREGr trials to increase constantly from block to block, in contrast to the slow plateau shown in the behavioural data. It seems likely, therefore, that there remains a non-trivial ‘fixing’ effect that may reflect consolidation processes, not accounted for by the current model (to our knowledge there is no other statistical learning model that accounts both for learning dynamics and long-term fixed effects).” (subsection “Modelling”).

“The persistence of a stable RT advantage 24 hours and 7 weeks after initial exposure demonstrates the establishment of a long-term memory representation, possibly through a process of consolidation involving long-lasting synaptic changes (Phan et al., 2017; Redondo and Morris, 2011). It may also be tempting to interpret the resistance to interruption, observed in early stages of memory formation (Exp. 3, Exp. S2), as a hint that a form of consolidation might have occurred already after a few initial presentations.” (subsection “Time scales of memory for sequences”).